# Characterizing deep-water oxygen variability and seafloor community responses using a novel autonomous lander

Natalya D. Gallo[1,2,3], Kevin Hardy[4], Nicholas C. Wegner[2], Ashley Nicoll[1], Haleigh Yang[5], Lisa A. Levin[3,5]

[1]Marine Biology Research Division, Scripps Institution of Oceanography, University of California San Diego, La Jolla, California 92093, USA
[2]Fisheries Resources Division, Southwest Fisheries Science Center, NOAA Fisheries, 8901 La Jolla Shores Drive, La Jolla, CA 92037, USA
[3]Center for Marine Biodiversity and Conservation, Scripps Institution of Oceanography, University of California San Diego, La Jolla, California 92093, USA
[4]Global Ocean Design LLC, 7955 Silverton Avenue Suite 1208, San Diego, CA 92126, USA
[5]Integrative Oceanography Division, Scripps Institution of Oceanography, University of California San Diego, La Jolla, California 92093, USA

Correspondence to: Natalya D. Gallo (ndgallo@ucsd.edu)

**Abstract.** Studies on the impacts of climate change typically focus on changes to mean conditions. However, animals live in temporally variable environments that give rise to different exposure histories that can potentially affect their sensitivities to climate change. Ocean deoxygenation has been observed in nearshore, upper-slope depths in the Southern California Bight, but how these changes compare to the magnitude of natural $O_2$ variability experienced by seafloor communities at short timescales is largely unknown. We developed a low-cost and spatially flexible approach for studying nearshore, deep-sea ecosystems and monitoring deep-water oxygen variability and benthic community responses. Using a novel, autonomous hand-deployable Nanolander with an SBE MicroCAT and camera system, high-frequency environmental ($O_2$, T, estimated pH) and seafloor community data were collected at depths between 100-400 m off San Diego, CA to characterize: timescales of natural environmental variability, changes in $O_2$ variability with depth, and community responses to $O_2$ variability. Oxygen variability was strongly linked to tidal processes, and contrary to expectation, oxygen variability did not decline linearly with depth. Depths of 200 and 400 m showed especially high $O_2$ variability; these conditions may give rise to greater community resilience to deoxygenation stress by exposing animals to periods of reprieve during higher $O_2$ conditions and invoking physiological acclimation during low $O_2$ conditions at daily and weekly timescales. Despite experiencing high $O_2$ variability, seafloor communities showed limited responses to changing conditions at these shorter timescales. Over 5-month timescales, some differences in seafloor communities may have been related to seasonal changes in the $O_2$ regime. Overall, we found lower oxygen conditions to be associated with a transition from fish-dominated to invertebrate-dominated communities, suggesting this taxonomic shift may be a useful ecological indicator of hypoxia. Due to their small size and

ease of use with small boats, hand-deployable Nanolanders can serve as a powerful capacity-building tool in data-poor regions for characterizing environmental variability and examining seafloor community sensitivity to climate-driven changes.

## 1 Introduction

Natural environmental variability can affect the resilience or sensitivity of communities to climate change.
Communities and species living in variable environments are often more tolerant of extreme conditions than communities from environmentally stable areas (Bay and Palumbi 2014). For example, in seasonally hypoxic fjords, temporal oxygen variability influences seafloor community beta diversity patterns, and can allow certain species to live for periods of time under average oxygen conditions that are below their critical oxygen thresholds ($P_{crit}$) (Chu et al. 2018). In addition, the anthropogenic signal of deoxygenation takes a longer time to emerge in systems with higher natural oxygen variability
(Long et al. 2016, Henson et al. 2017), such as Eastern Boundary Upwelling systems. Natural variability of dissolved oxygen at different timescales is therefore an important environmental factor to consider when studying the impacts of deoxygenation on communities.

While data on shallow-water $O_2$ and pH variability have proven valuable for interpreting faunal exposures (Hofmann et al. 2011b, Frieder et al. 2012, Levin et al. 2015), high-frequency measurements are rare below inner shelf
depths. Specifically, datasets on organismal and community responses to environmental variability are rare for the deep sea; however, those that exist, are informative and illustrate a dynamic environment (Chu et al. 2018, Matabos et al. 2012). Studies from NEPTUNE (the North-East Pacific Time-Series Undersea Networked Experiments) in B.C. Canada show that even at 800-1000 m, fish behavior is linked to variations in environmental conditions across different temporal scales including day-night and internal tide temporalizations (Doya et al. 2014) and seasonal cycles (Juniper et al. 2013). Combined
high-frequency quantitative sampling of environmental and biological data allows examination of which processes shape benthic communities (Matabos et al. 2011, 2014).

Currently, tools for studying deep-water, seafloor ecosystems include deep-submergence vehicles (HOVs, AUVs, and ROVs), towed camera sleds, and trawls. These approaches typically require significant resource investment, and the use of large ships with winch capabilities. Moorings and cabled observatories are also very useful; however, these are usually
fixed to specific sites and are typically costly. Eddy correlation techniques are also used to measure non-invasive oxygen fluxes at the seafloor, however require ROVs or scuba divers to deploy (Berg et al. 2009).

Untethered instrumented seafloor platforms, sometimes called "ocean landers", have a long and rich history (Ewing and Vine 1938, Tengberg et al. 1995). These vehicles are self-buoyant, with an expendable descent anchor that is released by surface command or on-board timer, allowing the vehicle to float back to the surface. Autonomous landers have several
advantages for deep-sea research, such as lower cost combined with spatial flexibility. Unlike moorings or cabled

observatories, small landers (< 2 m high) can easily be recovered using small boats and redeployed to new depths and locations (Priede and Bagley 2000, Jamieson 2016).

This study focuses on oxygen variability and community composition at depths between 100-400 m along the nearshore environment in the Southern California Bight (SCB). This depth zone is of interest because it encompasses the oxygen limited zone (OLZ) ($O_2$ < 60 μmol $kg^{-1}$ as defined in Gilly et al. 2013) and supports many important recreational and commercial fish species, including many species of slope rockfish (genus *Sebastes*), which may be vulnerable to deoxygenation (Keller et al. 2015, McClatchie et al. 2010). The OLZ is a transition zone above the oxygen minimum zone (OMZ, $O_2$ < 22.5 μmol $kg^{-1}$), where dissolved oxygen levels exclude hypoxia-intolerant species. The shallow upper OMZ boundaries likely experience more temporal variability than lower boundaries because upwelling is a generally shallow phenomenon (< 200 m) and these waters may be more biogeochemically responsive to changes in surface production. Thus, seafloor communities in the OLZ may be highly responsive to short-term and seasonal changes in oxygenation.

Upper slope depths on the US West Coast appear to be especially affected by global trends of oxygen loss (Levin 2018) and long-term trends have been captured by the California Cooperative Oceanic Fisheries Investigations (CalCOFI) quarterly measurements over the past 70 years. Off Monterey Bay in Central California, depths between 100-350 m have seen declines in oxygen of 1.92 μmol $kg^{-1}$ $year^{-1}$ between 1998-2013 (Ren et al. 2018). In the SCB, oxygen declines of 1-2 μmol $kg^{-1}$ $year^{-1}$ have been reported by several studies over a period of ~30 years (Bograd et al. 2008, 2015, McClatchie et al. 2010, Meinvielle and Johnson 2013), with the largest relative changes occurring at 300 m (Bograd et al. 2008). The proposed mechanisms for observed oxygen loss include increased advection of and decreasing oxygen in Pacific Equatorial Water (PEW) (Meinvielle and Johnson 2013, Bograd et al. 2015, Ren et al. 2018) and increased respiration, which is suspected to contribute more to oxygen loss at shallower depths (< 150 m) (Booth et al. 2014, Bograd et al. 2015, Ren et al. 2018). Long-term declines in pH and aragonite saturation state have also been demonstrated for the California Current System over similar periods using ROMS simulations (Hauri et al. 2013). Understanding the superposition of these long-term trends on natural high-frequency variability will be key to evaluating biotic responses.

Despite well documented long-term trends, no published high-frequency deep-water $O_2$ or pH measurements are available for depths below 100 m in the Southern California Bight (SCB). Notably, daily, weekly and even seasonal low oxygen extreme events (e.g. Send and Nam 2012) are not captured by the quarterly CalCOFI sampling frequency. Due to the physical oceanography and variable bathymetry of the SCB, nearshore deep-water areas on the shelf and slope are thought to experience high variability due to localized wind-driven upwelling events (Send and Nam 2012) and mixing from internal waves (Nam and Send 2011). In addition, the California Undercurrent transports warm, saline, low-oxygen subtropical water northward along the coast in the SCB and varies seasonally in strength, depth, and direction (Lynn and Simpson 1987), likely contributing to deep-water $O_2$ variability.

The goals of this study were to: (i) increase sensor accessibility to nearshore deep-water ecosystems through the development and testing of a small autonomous Nanolander, (ii) characterize deep-water $O_2$ variability over hours, days, and weeks at upper slope depths in the SCB relative to mean deoxygenation trends over decades, and identify dominant

timescales and depths of variability between 100-400 m, (iii) describe shelf and slope assemblages using the Nanolander camera system, and (iv) examine if and how the seafloor community responds to $O_2$ variability at short timescales (daily, weekly, seasonal) in terms of community composition and diversity.

## 2 Methods

### 2.1 Nanolander development and deployment

Autonomous landers have been used successfully to observe abyssal and deep-sea trench communities (e.g., Jamieson et al. 2011, Gallo et al. 2015), however, these landers were large and required a ship with an A-frame and winch to deploy and recover. For this study, the goal was to develop a deep-water lander that could easily be hand-deployed out of a small boat and that was capable of continuously collecting hydrographic and fish and invertebrate assemblage data from near the seafloor for several weeks at a time. With this goal in mind, the "Nanolander" *Deep Ocean Vehicle (DOV) BEEBE*, was
developed and built (Global Ocean Design, San Diego, CA) (Fig. 1A, Fig. 2). *DOV BEEBE* is named for William Beebe (1877-1962) who illuminated the deep-sea world during his Bathysphere dives (Beebe 1934).

      *DOV BEEBE* stands 1.6 m tall and is 0.36 m wide and 0.36 m deep. When *DOV BEEBE* is deployed, the vertical distance from the base of the Nanolander to the seafloor is ~51 cm (Fig. 1B). This distance is defined by the length of the anchor chain connecting the lander release system to the expendable iron anchor and may be shortened or lengthened. The
Nanolander frame is made of marine-grade high-density polyethylene (HDPE) (brand name "Starboard") and reinforced with fiberglass pultruded channel and angle beams for structure, reducing in-water weight. HDPE has a specific gravity of <1, close to neutrally buoyant. The specific gravity of fiberglass is 2/3 that of aluminium, requiring less flotation to achieve neutral buoyancy. Both plastic materials are impervious to saltwater corrosion. Alloy 316 stainless steel fasteners hold the frame together.

Within the frame sit three plastic spheres that are 25.4 cm in diameter; the spheres are made of injection-molded polyamide with 15% glass fibers for additional strength. The novel design aspects of the Nanolander include the use of plastic spheres for both instrument housing and flotation, which allow the vehicle to be smaller and lighter. Previous generation landers, such as the landers used for the *DEEPSEA CHALLENGE* Expedition (Gallo et al. 2015), used syntactic foam for flotation, which is more expensive and requires a metal support frame. While glass spheres have a deeper
maximum operational depth (to 11 km), the use of glass-filled polyamide spheres in the Nanolander has machining advantages and decreases the price point. The plastic spheres used for *DOV BEEBE* are pressure-tolerant to 1 km; new spheres with 20% glass content are pressure tolerant to 2 km.

      All three main spheres of *DOV BEEBE* are used to support electronics and instrumentation required for deployment, data collection, and recovery. The upper sphere houses an Edgetech BART (Burnwire-Acoustic Release-Transponder) board,
which is the prime means of communication with the Nanolander. A transducer is bonded to the exterior of the upper sphere, positioned to point upwards with a clear view to the surface. The Edgetech BART board has four pre-programmed

commands that enable and disable acoustic responses, and initiate the burn command for recovery. An overboard transducer and an Edgetech deckbox are used to communicate acoustically with the upper sphere, and allow distance ranging on the Nanolander. The battery for the BART board is housed in the upper sphere with the BART board.


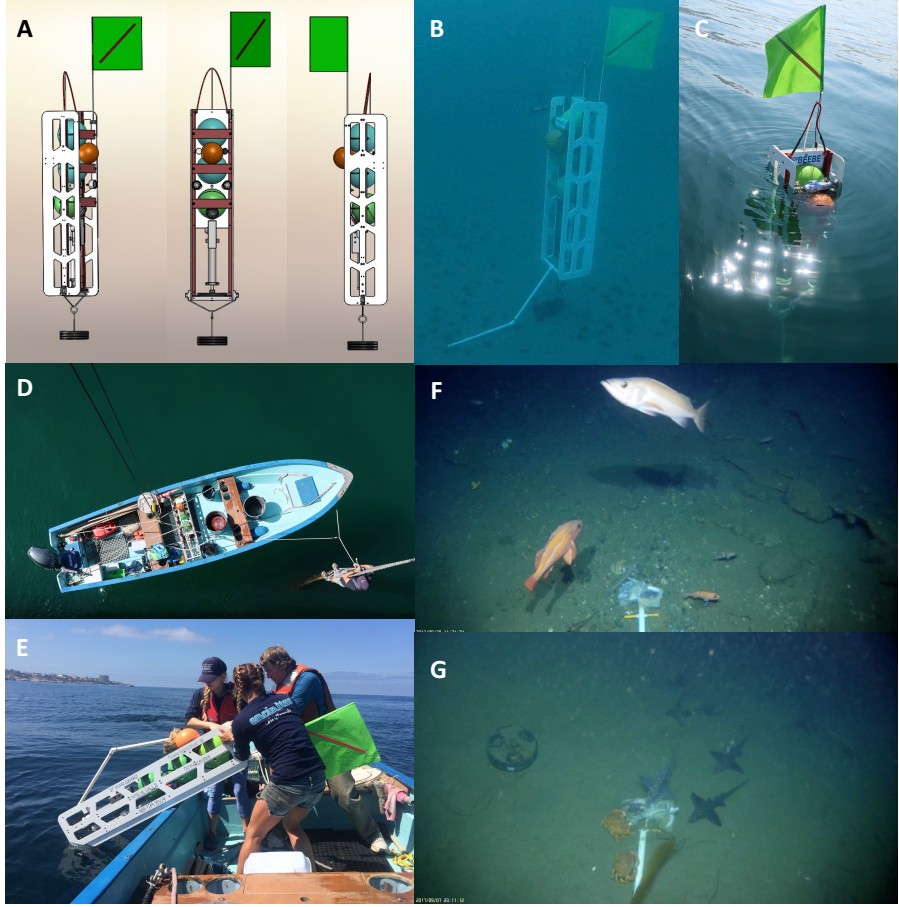

**Figure 1:** ***DOV BEEBE*** **is an autonomous, hand-deployable Nanolander capable of operating to 1000 m depth. It is outfitted with a Seabird MicroCAT-ODO environmental sensor for collecting high-frequency measurements of near-seafloor temperature, oxygen, salinity, and pressure, and a camera and light system for collecting videos of seafloor communities. (A) Front and side views**
**showing the general design for** ***DOV BEEBE*** **(see additional details in Fig. 2).** ***DOV BEEBE*** **is shown deployed at 30 m depth in (B), and floating at the surface in (C) prior to recovery.** ***BEEBE*** **can easily be deployed and recovered by hand from a small boat by as few as two people (D, E). Examples of the field of view from the** ***BEEBE*** **camera system taken from (F) D100-DM-Fall at ~100 m off Del Mar Steeples Reef showing various rockfish species (genus** *Sebastes***) and (G) D200-LJ-2 at ~200 m near the Scripps Reserve showing the presence of cancer crabs and chimaeras. The drop arm with the bait bag can be seen in camera field of view.**

The middle sphere functions as a "battery pod" and houses the batteries and battery management system (BMS) that power the two external LED lights. The LED lights are powered by a circuit consisting of five components: Battery > BMS > Relay > LED Driver > LED lights. Rechargeable lithium-polymer batteries were used and can be recharged through an external cable and charger system without having to open the middle sphere. For all but one deployment, a 30-ampere hour

(Ah) battery stack was used to power the LED lights, which was composed of three 14.8v/10 Ah units. For the last
deployment, power capacity was upgraded to 32 Ah by using two 14.8v/16 Ah batteries. In each case, each individual
battery (10 or 16 Ah) had its own battery management system (BMS) with a low voltage cut-out (LVCO) to ensure that
battery discharge never went below a critical threshold (12.0 v), which would damage the battery. This battery system is
novel and was developed specifically to fit within the spatial constraints of the sphere and provide high power capacity.

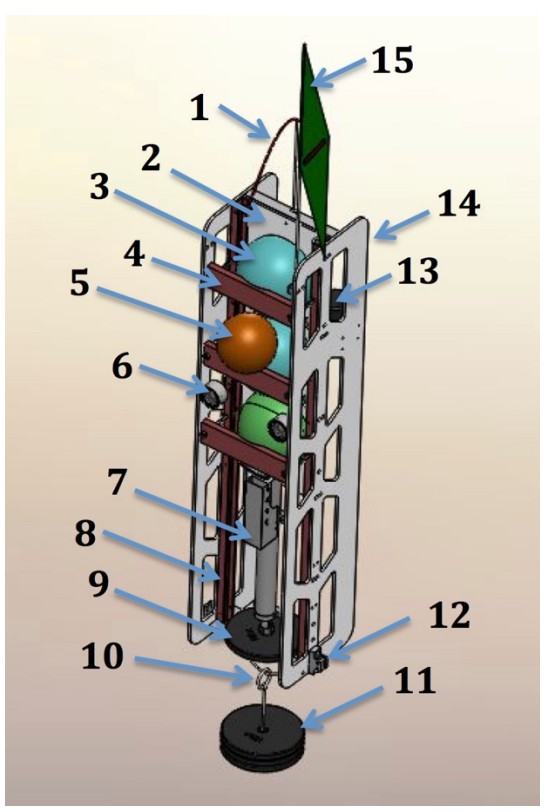


**Figure 2: A detailed schematic of the Nanolander *DOV BEEBE* components: 1) Spectra lifting bale; 2) High-density polyethylene (HDPE) centerplate; 3) ~25 cm polyamide spheres stacked top, middle and bottom, see description, Section 2.1; 4) sphere retainer; 5) auxiliary ~18 cm flotation sphere; 6) oil-filled LED lights; 7) Seabird MicroCAT-ODO in the lower payload bay; 8) central fiberglass frame; 9) stabilizing counterweight; 10) anchor slip ring; 11) expendable iron anchor (bar bell weights); 12) burnwire**
**release and mount, one port side, one starboard side; 13) Edgetech hydrophone for acoustic command and tracking; 14) HDPE side panels; and 15) surface recovery flag. Not shown: drop arm on front (see Fig. 1B and 1E).**

The relay and LED drivers are contained within the lower sphere, which houses all components of the camera
system, and includes the viewport. The camera system uses a Mobius Action Camera with a time-lapse assembly, which was
modified by Ronan Gray (SubAqua Imaging Systems, San Diego, CA) and William Hagey (Pisces Design, La Jolla, CA).
The SphereCam manual is provided as Supplement 1A. The camera system has 14 different time-lapse options, including
continuous video, time-lapse images, and time-lapse video at pre-programmed intervals. For these deployments, a sampling
interval of 20 seconds of video every 20 minutes was used. A sealed magnetic switch triggers the camera system to begin

programmed sampling at a pre-determined interval. An internal light-sensitive relay, pointed towards a camera indicator LED, triggers the external LED lights to power on. The two LED lights are attached to the body of the Nanolander and positioned on either side of the middle sphere. All spheres were sealed with a Deck Purge Box (Global Ocean Design, San Diego, CA) using a desiccant cartridge to remove moisture, and were held together by a vacuum of ~7 psi (~1/2 atm).

Below the bottom camera sphere, *DOV BEEBE* has a mounted SBE 37-SMP-ODO instrument (Sea-Bird Scientific) with titanium housing, rated to 7000 m. The MicroCAT CT(D)-DO is a highly accurate sensor designed for moorings and other long-duration, fixed-site deployments. It includes a conductivity, temperature, pressure, and SBE 63 optical dissolved oxygen sensor. Initial sensor accuracy is +/- 3.0 $\mu$mol kg$^{-1}$ for oxygen measurements, +/- 0.1% for pressure measurements, +/- 0.002°C for temperature measurements, and +/- 0.0003 S m$^{-1}$ for conductivity measurements, and drift is minimal. The SBE MicroCAT was programmed to take samples every five minutes for the length of the whole deployment.

A drop-arm is mounted on the front of *DOV BEEBE*, and is secured with a release during deployment. Initially a galvanic release was used, but a stack of 3-4 "Wint O Green" lifesavers was found to be more time-efficient. The drop-arm served three functions: it stabilized the Nanolander when exposed to current, it had a ~15 cm cross-bar for visual sizing reference, and it was used to attach bait for each deployment. The bait used was composed of an assortment of previously frozen demersal fishes that are part of the SCB upper margin demersal fish community. Bait was secured within a mesh cantaloupe bag and secured to the drop-arm with Zip-Ties for each deployment. All bait had been eaten by recovery.

*DOV BEEBE* is positively buoyant in water, and is deployed with ~18 kg of sacrificial iron weights. The weights are attached by a sliding link onto a metal chain, which is secured on each side to the base of the Nanolander using a burn wire (Fig. 2). Successful release of either burn wire allows the metal link to slide off the chain and drop the weights, releasing the Nanolander from the bottom. *DOV BEEBE*'s estimated descent rate is ~100 m per minute, and ascent rate is ~60 m per minute, following release of the weights. Once at the surface, *DOV BEEBE* floats ~0.45 m above the water and has a large flag, which assists with visual detection of the Nanolander (Fig. 1C). In addition to the three main spheres, additional smaller spheres were used, as needed, to increase buoyancy (Fig. 2).

Seven deployments were conducted during the study period, ranging from 15-35 days, and at targeted depths of 100-400 m (Table 1). Two early deployments (D200-LJ-1 and D200-LJ-2) were done near the Scripps Reserve off La Jolla, CA and five subsequent deployments (D100-DM-Fall, D200-DM, D300-DM, D400-DM, D100-DM-Spr) targeted a nearby rockfish habitat – the Del Mar Steeples Reef, CA (Fig. 3). Despite relatively close spatial proximity (~10 km), the local bathymetry differed between the LJ and DM deployments (Fig. 3); the LJ deployments were close to a submarine canyon feature, while the DM deployments were on a gradually sloping margin. Environmental and camera-based community data were collected during six of the seven deployments; only environmental data are available from the first deployment (D200-LJ-1) due to a camera technical problem. We aimed to conduct repeat deployments at each site to capture seasonal differences between a period of relaxed upwelling (fall/winter) and a period of strong upwelling (spring/summer), however full sampling was not feasible due to time and equipment constraints. Consequently, only one repeat deployment is available for ~100 m at Del Mar Steeples Reef (D100-DM-Fall and D100-DM-Spr).

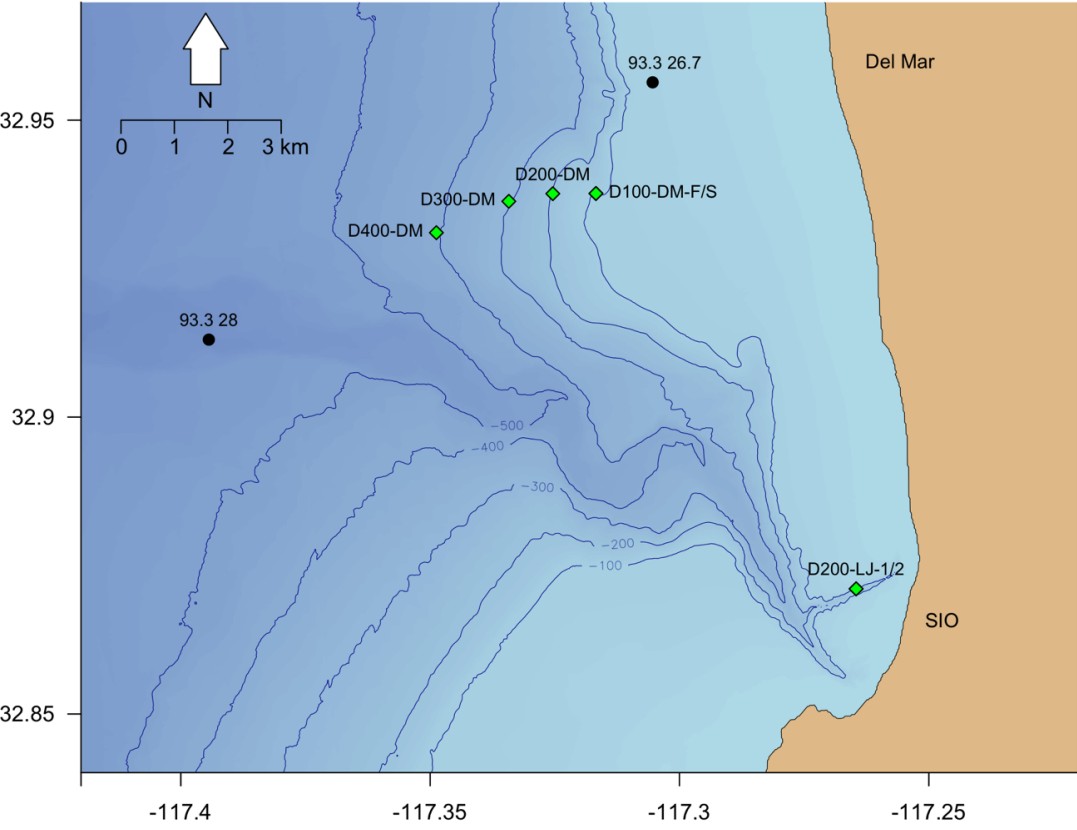

**Figure 3: Map of Nanolander *DOV BEEBE* deployments shown in relation to local bathymetry and nearby stations sampled quarterly by the California Cooperative Oceanic Fisheries Investigations (CalCOFI). Green diamonds indicate Nanolander deployments, black circles indicate CalCOFI stations with station labels, and isobaths show 100, 200, 300, 400, and 500 m depth contours. Note that green diamonds labelled as D200-LJ-1/2 and D100-DM-F/S represent two deployments each, but points overlap due to proximity.**

### 2.2 Characterizing environmental variability on the shelf and upper slope

Upon recovery of the Nanolander, time-series data from the MicroCAT were analyzed to assess how environmental variability ($O_2$, T, salinity) changes with depth. Since partial pressure of oxygen may be more biologically meaningful than oxygen concentration for understanding animal exposures to oxygen, we also calculated oxygen partial pressure as in Hofmann et al. (2011a). Oxygen and pH naturally co-vary along the continental margin driven by respiration. To examine variability of carbonate chemistry parameters, pH, $\Omega_{arag}$, and $\Omega_{calc}$ were estimated using empirical equations derived for this region in Alin et al. (2012). $\Omega_{arag}$, and $\Omega_{calc}$ are the calcium carbonate saturation state of aragonite and calcite respectively, and conditions favor calcium carbonate dissolution when $\Omega < 1$.

The mean and ranges of environmental conditions were compared across depths and deployments to characterize differences in environmental variability that seafloor communities were exposed to over short timescales. Probability density

distributions of environmental conditions were used to visualize differences in environmental conditions for each deployment. The coefficient of variation (CV) (i.e. the ratio of the standard deviation to the mean) was calculated for

environmental variables for each deployment as a standardized measure of dispersion and compared across deployments and depths. Additionally, the percent of measurements in which conditions were hypoxic ($O_2$ < 60 µmol kg$^{-1}$), severely hypoxic ($O_2$ < 22.5 µmol kg$^{-1}$), or undersaturated with respect to aragonite ($\Omega_{arag}$ < 1) or calcite ($\Omega_{calc}$ < 1) was determined for each deployment.

**Table 1: Information for seven deployments conducted with the Nanolander *DOV BEEBE* including deployment dates, length, location, depth, environmental conditions for each deployment, total number of 20-second video samples available for the community analysis, and camera and light performance for each data deployment. [$O_2$] < 60 µmol kg$^{-1}$ is defined as hypoxic, [$O_2$] < 22.5 µmol kg$^{-1}$ is defined as severely hypoxic, and $\Omega_{arag}$ < 1 and $\Omega_{calc}$ < 1 are defined as undersaturated with respect to aragonite or calcite, respectively. Mean and range $O_2$ percent saturation and p$O_2$ (kPa) for each deployment are provided in Supplement 1B.**
**CV = coefficient of variation (i.e. the ratio of the standard deviation to the mean). pH$_{est}$ is estimated pH, calculated using empirical relationships from Alin et al. (2012).**

| | D200-LJ-1 | D200-LJ-2 | D100-DM-Fall | D200-DM | D300-DM | D400-DM | D100-DM-Spr |
|---|---|---|---|---|---|---|---|
| **Dates** | Aug 17-Sep 1, 2017 | Sep 7-Sep 25, 2017 | Sep 29-Nov 3, 2017 | Nov 9-29, 2017 | Dec 12, 2017 - Jan 5, 2018 | Jan 23-Feb 8, 2018 | Mar 8-Mar 29, 2018 |
| **Deployment Length** | ~15 days | ~19 days | ~35 days | ~20 days | ~24 days | ~16 days | ~21 days |
| **Location** | Scripps Coastal Reserve (32.87108° N, 117.26459° W) | Scripps Coastal Reserve (32.87108° N, 117.26457° W) | Del Mar Steeples Reef (32.93765° N, 117.31675° W) | Del Mar Steeples Reef (32.93762° N, 117.3254° W) | Del Mar Steeples Reef (32.93633°N, 117.33422°W) | Del Mar Steeples Reef (32.93105°N, 117.34875°W) | Del Mar Steeples Reef (32.93765°N, 117.31675°W) |
| **Bottom Depth (m)** | 179 | 178 | 99 | 192 | 295 | 399 | 98 |
| **Mean Temp (°C)** | 10.07 | 9.88 | 11.10 | 9.51 | 8.39 | 7.42 | 9.80 |
| **Temp Range (°C)** | 9.72-10.43 | 9.45-10.44 | 10.35-12.26 | 8.94-10.21 | 7.99-8.77 | 6.97-7.89 | 9.39-10.30 |
| **CV Temp (%)** | 1.35 | 1.69 | 2.89 | 2.11 | 1.88 | 2.02 | 1.70 |
| **Mean [$O_2$] (µmol kg$^{-1}$)** | 70.75 | 77.61 | 132.00 | 82.10 | 49.38 | 28.97 | 103.95 |
| **[$O_2$] range (µmol kg$^{-1}$)** | 48.82-103.87 | 49.41-108.26 | 110.40-156.50 | 63.33-102.96 | 39.89-59.36 | 21.19-38.41 | 91.22-123.01 |
| **CV [$O_2$] (%)** | 13.72 | 12.92 | 5.07 | 9.82 | 7.02 | 10.20 | 5.72 |
| **Mean pHest** | 7.646 | 7.655 | 7.759 | 7.658 | 7.594 | 7.553 | 7.70 |
| **pHest Range** | 7.607-7.704 | 7.605-7.711 | 7.713-7.814 | 7.625-7.699 | 7.575-7.613 | 7.538-7.572 | 7.671-7.732 |
| **CV pHest (%)** | 0.22 | 0.23 | 0.2 | 0.19 | 0.09 | 0.08 | 0.15 |
| **Conditions hypoxic (% time)** | 12.64% | 1.88% | 0% | 0% | 100% | 100% | 0% |
| **Conditions severely hypoxic (% time)** | 0% | 0% | 0% | 0% | 0% | 1.12% | 0% |
| **Conditions undersaturated (aragonite) (% time)** | 99.65% | 99.77% | 0% | 100% | 100% | 100% | 92.80% |
| **Conditions undersaturated (calcite) (% time)** | 0% | 0% | 0% | 0% | 0% | 97.30% | 0% |
| **Number of 20-sec video samples for analysis** | N/A | 1009 | 876 | 1012 | 406 | 594 | 396 |
| **Number of video samples with good visibility** | N/A | 1009 | 876 | 656 | 6 | 594 | 396 |
| **Amount of time before lights first failed (h)** | N/A | 5.61 | 4.77 | 5.63 | 2.26 | 3.30 | 2.21 |

Previous studies have found that changes in oxygen and pH in the Southern California Bight are associated with changes in the volume of Pacific Equatorial Water (PEW) transported in the California Undercurrent (Bograd et al. 2015, Nam et al. 2015). PEW is characterized by low oxygen, warm, high salinity conditions, and is composed of two water masses, the 13°C water mass (13CW) and the deeper Northern Equatorial Pacific Intermediate Water Mass (NEPIW) (Evans et al. 2020). Spiciness, the degree to which water is warm and salty, is a state variable that is conserved along isopycnal surfaces (Flament 2002) and can be used as a tracer for PEW (Nam et al. 2015). We calculated spiciness using the "oce" R package (Kelley and Richards 2017) and examined how oxygen concentration varies with temperature and spiciness across depths and deployments. Spiciness is used to examine differences in spatial variation between water masses, which otherwise may not be apparent using isopycnal surfaces because the effects of warm temperature and high salinity cancel each other out. "Spicier" water is warmer and saltier.

To identify the dominant timescale of variability for oxygen, a spectral analysis was conducted as in Frieder et al. (2012) on the oxygen time series for each deployment. To look at diurnal and semidiurnal patterns, one day was used as the unit of time, and the number of observations based on the sampling frequency, was 288. Spectral analyses were conducted on a detrended time series using a fast fourier transform. Results were displayed using a periodogram and the period of the dominant signal was compared across deployments. The oxygen time series for each deployment was also decomposed using the "stats" package (R Core Team 2019) to look at the trend, daily, and random signals that contribute to the overall data patterns.

**2.3 Short-term oxygen variability in the context of longer trends**

To examine $O_2$ variability over shorter (i.e. daily and weekly) timescales compared to longer (i.e. seasonal, interannual, multidecadal) timescales, we compared our Nanolander results with the annual rates of oxygen loss reported for the SCB nearshore region (Bograd et al. 2008) as well as CTD casts from nearby CalCOFI station 93.3 28 (Fig. 3). CalCOFI station 93.3 26.7 was also nearby, but was too shallow for comparison with the Nanolander deployments (Fig. 3). Quality controlled CTD casts from station 93.3 28 were available for a ~16-year period (Oct 2003-November 2019), representing data from 61 cruises (calcofi.org). CTD data were used to examine characteristics of variability of temperature and oxygen through the water column, including mean conditions, the standard deviation, and the coefficient of variation across the 16-year period of quarterly samples. Oxygen data at 100, 200, 300, and 400 m were extracted to compare the distribution of observations across this 16-year period with the high-frequency measurements from the ~3-week Nanolander deployments. Additionally, we tested for significant linear trends in temperature or oxygen at 100, 200, 300, and 400 m, to examine recent (2003-2019) warming and deoxygenation trends at the CalCOFI station closest to the Nanolander deployments.

**2.4 Assessing community responses to oxygen variability**

Video segments recorded by the camera system were annotated to analyze if and how seafloor communities differ with respect to environmental conditions. A total of 4,293 20-second video segments were collected and annotated in total.

For each 20-second video, both invertebrates and vertebrates within the frame of view were identified to lowest taxonomic level and counted.

    Since visibility was impaired during certain deployments due to high turbidity, each video clip was categorized by visibility quality using the following categories: 1 (can see the bottom, good visibility), 2 (can only see the drop-arm, poor visibility), or 3 (drop-arm can no longer be seen, no visibility). Only samples with a visibility category of 1 were utilized in

subsequent community analyses so that differences in community patterns were not due to differences in visibility.

    Non-metric multidimensional scaling was used to assess community-level differences across deployments. The R package "vegan" (Oksanen et al. 2017) was used for nMDS analysis and a Wisconsin double standardization was performed and counts were transformed using a square-root transformation. These standardizations are frequently used when working with datasets with high-count values and have been found to improve nMDS results (Oksanen et al. 2017). Bray-Curtis

dissimilarity was used as the input and community dissimilarities were mapped onto ordination space for the nMDS analysis. Rare species (<8 observations across all deployment samples) and video samples with only one animal observation were removed from the community matrix, resulting in a total number of 3357 video samples and 43 unique species included in the community analysis.

    Since fishes are typically less hypoxia-tolerant than invertebrates (Vaquer-Sunyer and Duarte 2008), we

hypothesized there would be a shift from a fish- to an invertebrate-dominated seafloor community that correlated with decreasing oxygen conditions. Samples from all deployments were categorized as "Fish Dominant", "Equal", or "Invertebrate Dominant" based on if there were more fishes or more invertebrates observed in each 20-second video sample. These categories were then projected onto ordination space and superimposed with oxygen contours using the ordisurf function in "vegan".

Since low oxygen conditions have been found to depress fish diversity (Gallo and Levin 2016), fish species accumulation curves relative to the number of video samples were examined to look at differences in fish diversity across deployments. We selected this metric of diversity since the number of video samples differed across deployments (Table 1). Only video samples in which fish were present were included in the calculation of the species accumulation curves. A table of all fish species observed during the deployments is included in Supplement 1C.

To test the ability of the Nanolander to capture short-term responses in seafloor communities, we selected two deployments that had high environmental variability (D200-LJ-2 and D200-DM). For these deployments, samples were grouped in day (6:00 am-5:59 pm PST) and night (6:00 pm-5:59 am PST) categories and oxygen categories ("High", "Intermediate", and "Low"). Oxygen categories were determined separately for each deployment based on the deployment time series, and were selected to showcase extremes: "High" samples represented the highest 10% of observed oxygen

conditions, and "Low" samples the lowest 10% of observed oxygen conditions for the deployment. All other samples were categorized as "Intermediate." An nMDS analysis was performed to look at differences in communities in relation to diurnal patterns and oxygen conditions within the timeframe of a single deployment. Rare species with fewer than three observations across the deployment time series were removed from the community matrices, resulting in a community matrix with 844

video samples and 19 species for D200-LJ-1 and 645 video samples and 17 species for D200-DM. These were used in the nMDS analysis.

## 3 Results

### 3.1 Nanolander performance

*DOV BEEBE* was found to be a reliable platform for deployment, recovery, and data collection. Small boats were used for deployment and recovery (Fig. 1D and E) and *DOV BEEBE* was easily transported by lab cart or car. The Nanolander framework was robust and showed very few signs of wear following multiple deployments. Spheres showed no signs of leakage or vacuum loss, and acoustic communication worked well during all deployments.

Memory and power capacity often limit deployment times for long-term, deep-sea deployments. In this study, the main technological limitation we ran into was limited battery capacity to power the LED lights. As opposed to 8 hours of estimated LED performance time, field performance ranged from 2.2 to 6.6 hours total time, which meant that the total time of biological data collection was shortened and ranged from 5.5 to 16.5 days, respectively (Table 1). Memory and power were not issues for the camera system; the 128 GB micro SD card was cleared and the battery pack was fully recharged following each deployment. Video quality was high enough to allow species-level identifications and the light from the LEDs was sufficient to light the field of view (Fig. 1F and G). The SeaBird MicroCAT-ODO also performed without any issues and had sufficient battery and memory capacity for all deployments. If not for power limitations to the LED lights, the camera system and SBE MicroCAT would have allowed for longer sampling (~1 month and potentially longer). The basic Nanolander itself can stay *in situ* for up to two years. Detailed descriptions of Nanolander performance can be found in Gallo (2018).

### 3.2 Characteristics and drivers of oxygen variability across short time-scales

Natural variability of environmental parameters was assessed from time series data collected during each deployment and compared across depths (100, 200, 300, and 400 m), and season (fall compared to spring). Means and ranges for temperature, oxygen, salinity, and $pH_{est}$ for each deployment were determined (Table 1). At ~100 m, conditions were never hypoxic (i.e. < 60 μmol kg$^{-1}$), although the mean oxygen concentration was significantly lower during the spring upwelling season deployment (D100-DM-Spr, mean $O_2$ = 104 μmol kg$^{-1}$), compared to the fall deployment when upwelling was relaxed (D100-DM-Fall, mean $O_2$ = 132 μmol kg$^{-1}$) (ANOVA, p < 0.001). $pH_{est}$ was also lower during the spring deployment (D100-DM-Spr, mean $pH_{est}$ = 7.696) than during the fall deployment at ~100 m (D100-DM-Fall, mean $pH_{est}$ = 7.759) (ANOVA, p < 0.001), and temperatures were on average 1.3°C colder, consistent with upwelling conditions (Table 1, Fig. 4). While conditions were never undersaturated with respect to aragonite ($\Omega_{arag}$ < 1) during the fall deployment, during the spring deployment, conditions were undersaturated ~93% of the time (Table 1).

At ~200 m, hypoxic conditions ($O_2 < 60$ µmol kg$^{-1}$) were encountered, however conditions were only hypoxic for relatively short portions of the deployment (~13% for D200-LJ-1, ~2% for D200-LJ-2, and never hypoxic for D200-DM) (Table 1). Conditions were almost always undersaturated with respect to aragonite ($\Omega_{arag} < 1$) (Table 1). At ~300 m (D300-DM) and ~400 m (D400-DM), mean temperatures were colder than shallower depths, and mean oxygen and pH$_{est}$ conditions were lower than shallower depths (Table 1, Fig. 4). At both 300 and 400 m, conditions were continuously hypoxic, and at 400 m (D400-DM) conditions were severely hypoxic (i.e. $O_2 < 22.5$ µmol kg$^{-1}$) for ~1% of the time-series (Table 1). Both D300-DM and D400-DM were conducted during the fall/winter, when upwelling conditions are relaxed, therefore deployments likely captured the less extreme (higher oxygen, higher pH) conditions. At ~300 m, conditions were undersaturated with respect to aragonite ($\Omega_{arag} < 1$) but not calcite, whereas at ~400 m, conditions were also undersaturated with respect to calcite ($\Omega_{calc} < 1$) for most of the deployment (Table 1).

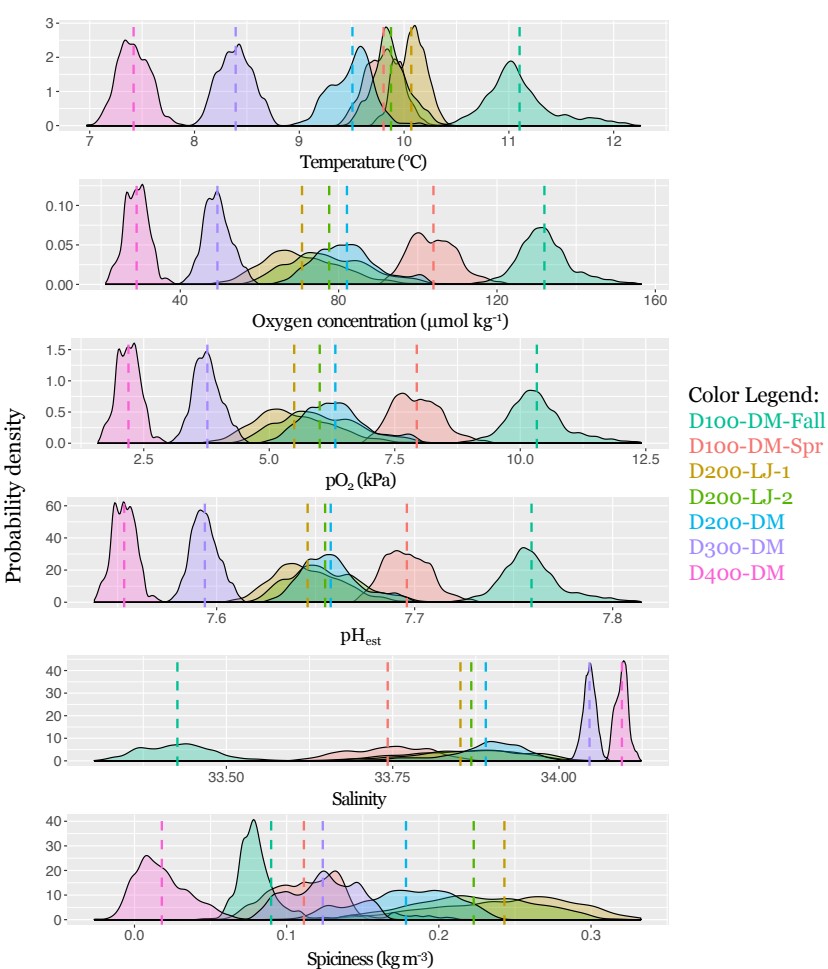

**Figure 4: Mean and variance of near-seafloor temperature, oxygen concentration, oxygen partial pressure, pH$_{est}$, salinity, and spiciness. The probability density of data collected for each deployment is shown, with the color of the data distributions corresponding to each deployment (as indicated in the color legend). The mean is indicated with a dotted line in the same color and**

**exact values are given in Table 1. pH$_{est}$ is estimated pH, calculated using empirical relationships from Alin et al. (2012). Sampling dates for each deployment are given in Table 1.**

While we expected that O$_2$ variability would decrease with depth, instead we found that the greatest variability in oxygen conditions over these short time-scales was observed at ~200 m (Table 1). All three deployments from ~200 m showed broad probability density distributions of environmental conditions (Fig. 4) and large ranges in oxygen and pH$_{est}$ for the deployment period (Table 1). The average daily range in oxygen concentration (i.e. daily maximum-daily minimum) was highest for D200-LJ-2 (~34 μmol kg$^{-1}$), followed by D200-LJ-1 (~31 μmol kg$^{-1}$), followed by D200-DM (~24 μmol kg$^{-1}$).

The average daily oxygen range for both ~100 m deployments was lower (~20 μmol kg$^{-1}$ for D100-DM-Fall and ~14 μmol kg$^{-1}$ for D100-DM-Spr). The coefficient of variation (CV) for oxygen at ~200 m was twice higher than for the ~100 m deployments (Table 1). While deployments at ~300 m (D300-DM) and ~400 m (D400-DM) had much narrower probability density distributions of environmental conditions (Fig. 4), the ranges in oxygen and pH$_{est}$ at ~400 m were only slightly smaller than at ~300 m (Table 1). The CV for oxygen was higher at ~400 m (10.20%) compared to ~300 m (7.02%) (Table 1).

1). The average daily range in oxygen concentration was ~11 μmol kg$^{-1}$ for D300-DM and ~8 μmol kg$^{-1}$ for D400-DM. Temperature did not exhibit the same pattern of variability as oxygen, with the highest variability (CV) observed during D100-DM-Fall (~100 m) (Table 1). Variability in pH$_{est}$ (CV) was almost twice higher at shallower depths (< 200 m), than at ~300 or ~400 m (Table 1).

    Using a spectral analysis, we found that the dominant frequency underlying oxygen variability for all deployments

was close to the semidiurnal tidal period (~12.4 hrs) (Supplement 1D). When the time series were decomposed into their additive components (i.e. daily trend, underlying trend, and random noise), time series for all depths showed a clear diurnal and semi-diurnal signal (Supplement 1E). Thus, oxygen variability on the outer shelf and upper slope is mainly driven by tides. The relative amplitude of the dominant signal in the periodogram decreases with increasing depth, suggesting that the strength of the tidal signal weakens with depth. Oxygen conditions tend to increase during ebb tide as the tide retreats, and

decrease during flood tide as the tide rises (Supplement 1F). Oxygen variability does not appear to increase with tidal amplitude; for D100-DM-Spr, D200-LJ-1, and D200-LJ-2 the daily oxygen range appears to be negatively correlated with the daily tidal range (Supplement 1F).

    Oxygen concentration was found to be significantly positively correlated with temperature for all deployments (LR, p < 0.001), however, the explanatory power of the regressions differed across depths (100, 200, 300, and 400 m) and the

slopes of the regressions differed between locations (Scripps Reserve and Del Mar Steeples Reef). At depths deeper than 200 m, there was less variance around the linear trend in oxygen. The highest amount of oxygen variance explained by the linear regression with temperature was found for D400-DM (~400 m, R$^2$ = 0.90), and the lowest amount for D200-DM (~200 m, R$^2$ = 0.41). The two deployments conducted near the Scripps Reserve (D200-LJ-1 and D200-LJ-2) had steeper slopes (Fig. 5) than deployments on the Del Mar Steeples Reef (D100-DM-Fall, D200-DM, D300-DM, D400-DM, D100-DM-Spr),

which may be related to bathymetric differences of the sites. Deployments near the Scripps Reserve were in a narrow, deep

tendril of the Scripps canyon system, which is surrounded by shallower bathymetry, while the Del Mar deployments were on a gradually sloping margin (Fig. 3).

Oxygen was significantly correlated with spiciness for all deployments (LR, $p < 0.001$), however, the slopes and explanatory power of this relationship differed across depths (100, 200, 300, and 400 m) and season (fall and spring) (Fig.

5). D100-DM-Fall and D100-DM-Spr were conducted at the same location at ~100 m, but during fall and spring, respectively, and exhibited differing relationships between oxygen and spiciness (Fig. 5). In the fall, the relationship between spiciness and oxygen at ~100 m was weak and positive with low explanatory power ($R^2 = 0.31$). In contrast, during spring, dissolved oxygen was negatively correlated with spiciness, and the linear fit had high explanatory power ($R^2 = 0.81$). At ~200 m (D200-LJ-1, D200-LJ-2, D200-DM), spiciness and oxygen concentration were also negatively correlated, with high

explanatory power for the linear fits ($R^2 = 0.98$, 0.92, and 0.61, respectively) (Fig. 5). At deeper depths (~300 and 400 m), the relationship between spiciness and oxygen was significant (LR, $p < 0.001$), but the correlation was positive with high explanatory power of the linear fit (D300-DM $R^2 = 0.61$, D400-DM $R^2 = 0.68$).

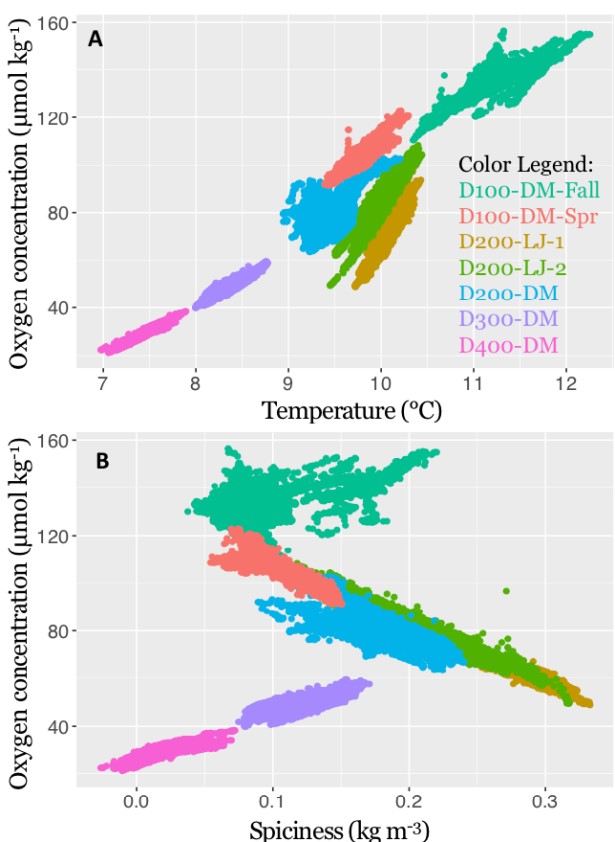

**Figure 5: 2017-2018 near-bottom dissolved oxygen concentration in the Southern California Bight shown in relation to**
**temperature (A) and spiciness (B). Data points represent samples taken every five minutes with the SBE MicroCAT-ODO sensor during the seven deployments. Deployments are distinguished by color, as indicated in the color legend. Sampling dates for each deployment are given in Table 1.**

### 3.3 Seafloor community differences and relationship to oxygen conditions

Community data were collected using the camera system during six deployments (Table 1), representing a total of
4,293 20-second videos that were annotated for organismal observations. Unexpected differences in visibility were observed across deployments. Clear conditions were present for deployments D200-LJ-2, D100-DM-Fall, D400-DM and D100-DM-Spr. During D200-DM at Del Mar Steeples Reef, visibility deteriorated throughout the deployment. The following deployment, D300-DM, which was at ~300 m at Del Mar Steeples Reef, had very poor visibility. For D300-DM, less than 2% of samples had good visibility, 78% had impaired visibility, and 20% had severely impaired visibility due to high sediment turbidity.

The community at the Del Mar Steeples Reef at ~100 m (D100-DM-Fall and D100-DM-Spr) was characterized by high numbers of rockfish (*Sebastes spp.*), especially halfbanded rockfish (*S. semicinctus*), but also included less-observed rockfish species such as the flag rockfish (*S. rubrivinctus*), bocaccio (*S. paucispinis*), rosy rockfish (*S. rosaceus*), and greenstriped rockfish (*S. elongatus*). Other commonly-observed fishes included the pink seaperch, *Zalembius rosaceus*, combfish, *Zaniolepis spp.*, and the spotted cusk-eel, *Chilara taylori*. Invertebrates were not abundant, but included an unidentified gastropod, the tuna crab, *Pleuroncodes planipes*, a yellow coral, as well as others. Except for the singular yellow coral, all other invertebrates were mobile. Seafloor communities for D100-DM-Fall and D100-DM-Spr were very similar, but were distinct from most other deployments (Fig. 6A).

Deployments D200-LJ-2 and D200-DM were in different locations (Table 1, Fig. 3), and the communities observed were very different (Fig. 6A) despite similar depth and environmental conditions (Fig. 4). Soft sediment characterized the benthos at both sites, but D200-LJ-2 was near a submarine canyon, while D200-DM was on a gradually sloping margin (Fig. 3). The community at D200-LJ-2 included eelpouts (*Lycodes spp.*), spotted cusk-eels (*C. taylori*), California lizardfish (*Synodus lucioceps*), and crabs (*Cancer spp.*), as well as, more typical deep-water species such as Dover sole (*Microstomus pacificus*), spotted ratfish (*Hydrolagus colliei*), and dogface witch eels (*Facciolella equatorialis*). In contrast, rockfish (*Sebastes spp.*), combfishes (*Zaniolepis spp.*), and Pacific sanddab (*Citharichthys sordidus*) were commonly observed during D200-DM, and the community was dominated by tuna crabs (*P. planipes*) and pink urchins (*Strongylocentrotus fragilis*) which were present in high abundances. Conversely, during D200-LJ-2, no pink urchins were observed, and tuna crabs were less abundant. Spot prawns (*Pandalus platyceros*) were common community members observed during both D200-LJ-2 and D200-DM, but were not observed during any other deployments.

Only one deployment was conducted at each of the two deeper depths (~300 m and 400 m), and both deployments were near the Del Mar Steeples Reef. Due to high turbidity, the bottom was only visible in a few samples from D300-DM. From these, it appeared that the community was dominated by tuna crabs (*P. planipes*) and pink urchins (*S. fragilis*), though in lower abundances than at D200-DM. Fish were rarely observed, but included Pacific hake (*Merluccius productus*), rockfish (*Sebastes spp.*), Pacific hagfish (*Eptatretus stoutii*), and hundred-fathom codling (*Physiculus rastrelliger*). D300-DM showed similarity to seafloor communities observed during D200-DM and D400-DM (Fig. 6A).

D400-DM represented the deepest deployment (~400 m) and had excellent visibility. The community was dominated by pink urchins (*S. fragilis*), but these were present in lower abundances than at D200-DM. Low numbers of tuna crabs (*P. planipes*) were also present. Fish were rare, but the fishes most commonly observed were Pacific hagfish (*E. stoutii*), blacktip poacher (*Xeneretmus latifrons*), dogface witch eels (*F. equatorialis*), Dover sole (*M. pacificus*), and shortspine thornyhead (*Sebastolobus alascanus*). Both fish and invertebrates were less active and showed less movement in D400-DM than at shallower deployments.

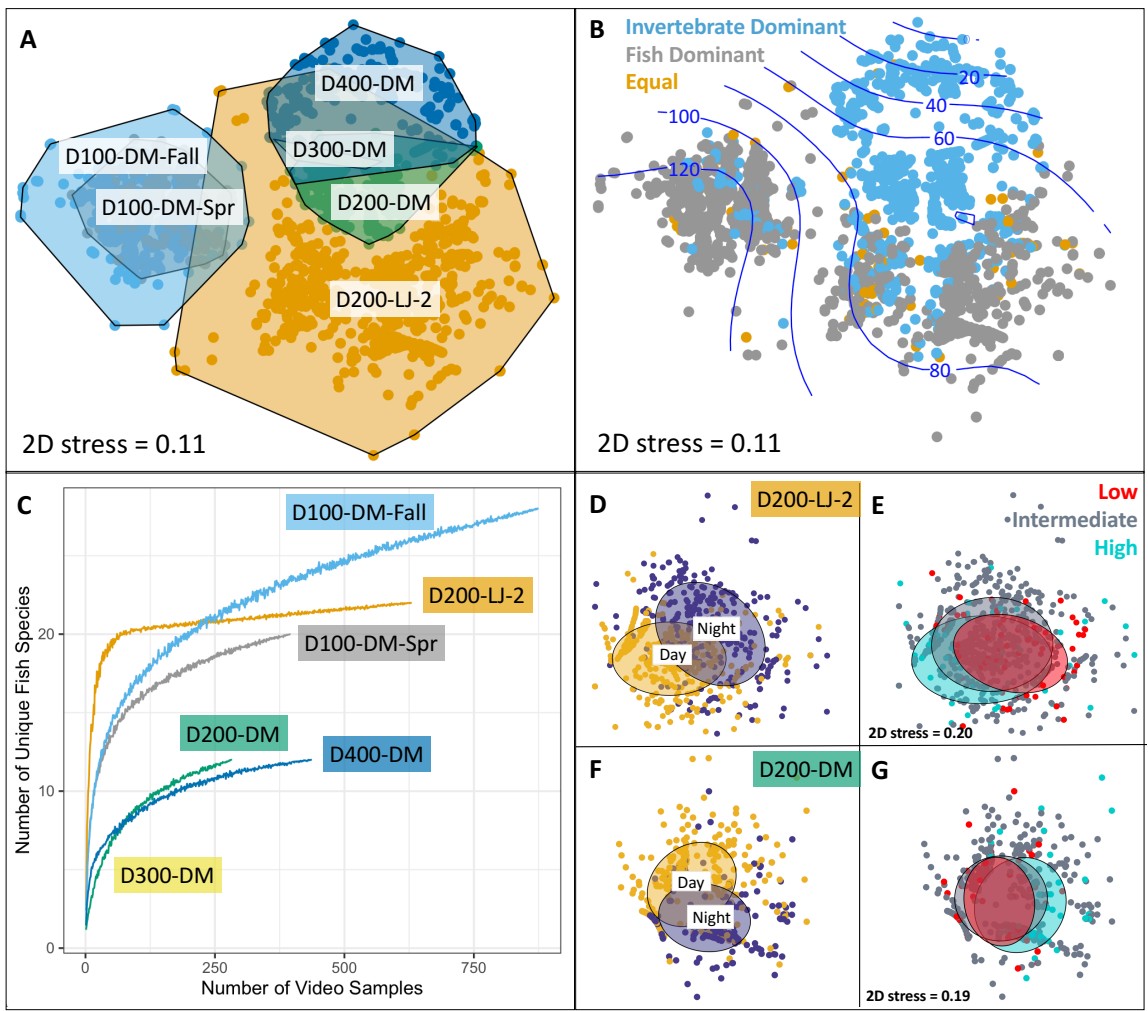

**Figure 6: Seafloor community analyses using *DOV BEEBE* video samples. A)** Non-metric multidimensional scaling (nMDS) plot showing seafloor community similarity across six deployments. Points represent Bray-Curtis similarity of square-root transformed counts of animals observed in each 20-second video sample (n = 3357) from each deployment (n = 6). Points are color-coded by deployment and a convex hull demarcates each deployment community. **B)** The same nMDS as in A) but points are color-coded by whether the seafloor community for each 20-second video sample was dominated by invertebrates (blue), vertebrates (gray), or an equal proportion of vertebrates and invertebrates (gold). Blue contours indicate relationship with oxygen concentration (μmol kg$^{-1}$). **C)** Species accumulation curves showing differences in fish diversity across deployments. **D-G)** Non-metric multidimensional scaling plots showing differences in seafloor community composition as a function of day versus night (D, F) and oxygen conditions

**(E, G) for two deployments: D200-LJ-2 (D, E) and D200-DM (F, G). In D) and F) yellow points represent daytime samples (6:00 AM – 5:59 PM) and purple points represent nighttime samples (6:00 PM – 5:59 AM). In E) and G), low (red) and high (blue) oxygen conditions represent the lowest and highest 10[th] percentile of oxygen conditions encountered during each deployment time series. Ellipses represent grouping by category and show 50% confidence limits. 2D stress is the same for D) and E) and for F) and G). See Table 1 for camera deployment details.**

We also looked at a community-level metric in relation to environmental oxygen conditions: community dominance by invertebrates or fishes. We hypothesized that higher-oxygen conditions would be characterized by fish dominance, compared to lower-oxygen conditions, which would be characterized by invertebrate dominance. Deployments D200-DM, D300-DM, and D400-DM were characterized by invertebrate dominance for either all or most (>98%) samples. In contrast, D200-LJ-2, D100-DM-Fall, and D100-DM-Spr were characterized by mixed communities, with fish-dominated communities more characteristic for D100-DM-Fall and D100-DM-Spr. In general, fish-dominated communities were more characteristic of higher-oxygen conditions when looking across all deployments (Fig. 6B), but we could not determine if this was specifically due to oxygen or other environmental covariates.

We were also able to examine differences in fish diversity across deployments using the Nanolander video samples. Species accumulations curves show differences in fish species diversity across deployments, with D100-DM-Fall having the highest number of observed fish species, followed by D200-LJ-2, D100-DM-Spr, and D200-DM and D400-DM which had the same number of unique fish species (Fig. 6C). The decline in fish diversity between D100-DM-Fall and D100-DM-Spr may be related to changes in environmental conditions between fall and spring, since the location is the same (Table 1).

Community-level changes within deployments were also examined for evidence of diurnal differences and differences related to oxygen concentration. D200-LJ-2 and D200-DM, which exhibited the highest oxygen variability and each had ~14-day time series of camera samples (Table 1, Fig. 6), were selected for further analysis. Clear diurnal differences were observed for both deployments (Fig. 6D, F), showing that at 200 m, communities are intimately linked to diurnal rhythms. Daytime communities were characterized by more combfishes (*Zaniolepis spp.*), hake (*M. productus*), small pelagic fishes such as the northern anchovy (*Engraulis mordax*), blacktip poachers (*X. latifrons*), and crabs (*Cancer spp.*), while nighttime communities were characterized by more lizardfish (*S. lucioceps*), spot prawns (*P. platyceros*), spotted ratfish (*H. colliei*), and hagfish (*E. stoutii*). Tuna crabs (*P. planipes*) and pink urchins (*S. fragilis*) showed no diurnal differences.

In contrast to clear diurnal differences, seafloor communities showed little evidence of responsiveness to changing oxygen conditions during the two deployments examined, however, some community-level differences do emerge when examining the highest and lowest oxygen conditions that were encountered during the deployment time series (Fig 6E, G). At ~200 m, crabs (*Cancer spp.*), spot prawns (*P. platyceros*), and lizardfish (*S. lucioceps*) were more common community members during the high oxygen extremes while tuna crabs (*P. planipes*) and Dover sole (*M. pacificus*) were more common during low oxygen extremes. For both deployments, the video time series lasted ~14 days, and a longer time series or more extreme oxygen variability may show more community-level differentiation in relation to oxygen extremes. Overall, our

results show that at short-timescales (2 weeks or less), seafloor communities responded to diurnal differences more than to high-frequency oxygen variability.

## 4 Discussion

The California Current System is expected to experience the impacts of hypoxia and ocean acidification on seafloor communities sooner than many other regions of the world (Alin et al. 2012) because upwelling brings deep, oxygen-poor,

and $CO_2$-rich waters into nearshore ecosystems along the US West Coast (Feely et al. 2008). Species in the SCB region may be particularly vulnerable to deoxygenation-induced habitat compression because the depth of the 22.5 μmol kg$^{-1}$ oxygen boundary (i.e. upper OMZ boundary) occurs at a shallower depth than in northern California, Oregon, and Washington (Helly and Levin 2004, Moffitt et al. 2015). This study shows that even during the relaxed upwelling season, seafloor communities at ~400 m can be periodically exposed to OMZ conditions, communities at ~300 m are continuously exposed to

hypoxic conditions, and communities at ~200 m are periodically exposed to hypoxic conditions. In the spring, upwelling of 13°C water (13CW) lowers oxygen conditions at 100 m, but conditions were never hypoxic in our study. Seafloor communities differed across the sampled environmental conditions, with communities living in lower-oxygen areas characterized by invertebrate dominance and decreased fish diversity.

### 4.1 Comparing oxygen variability: Short-term to long-term trends

Our Nanolander data show that at 100 m, benthic communities are exposed to ~4-7 μmol kg$^{-1}$ differences in oxygen conditions at semidiurnal timescales and ranges of 7-34 μmol kg$^{-1}$ at daily timescales (Supplement 1E, 1F). At ~200 m, benthic communities experienced higher oxygen variability of 10-12 μmol kg$^{-1}$ at semidiurnal timescales and ranges of 15-46 μmol kg$^{-1}$ at daily timescales (Supplement 1E, 1F). In contrast, semidiurnal and diurnal variability at 300 m and 400 m was reduced (Fig. 4, Table 1). At 300 and 400 m, a tidal signal still influenced oxygen conditions, but this signal was weaker and

oxygen varied only ~2 μmol kg$^{-1}$ at semidiurnal timescales (Supplement 1E). At daily timescales, oxygen varied between 8-15 μmol kg$^{-1}$ at 300 m, and 3-12 μmol kg$^{-1}$ at 400 m (Supplement 1F).

Across weekly timescales, at 100 m, oxygen conditions ranged by ~32 μmol kg$^{-1}$ during deployment D100-DM-Spr and 46 μmol kg$^{-1}$ during deployment D100-DM-Fall (Table 1). More extreme event-based decreases in oxygen have been reported near our study site at the Del Mar mooring (a continuous oceanographic monitoring mooring on the 100-m isobath;

Nam et al. 2015), but were not captured during any of our deployments. At ~200 m, the range in oxygen conditions across weekly timescales was similar or higher than at 100 m and was 55 μmol kg$^{-1}$ for D200-LJ-1, 59 μmol kg$^{-1}$ for D200-LJ-2, and 40 μmol kg$^{-1}$ for D200-DM. Similarly, the CV for oxygen was always higher during the 200 m deployments (13.72%, 12.92%, and 9.82%) compared to the 100 m deployments (5.07% and 5.72%) (Table 1). For the 300 m deployment, oxygen variability across weekly time-scales was lower than observed at ~200 m; the range in oxygen conditions was ~19 μmol kg$^{-1}$

and the CV was 7.02% (Table 1). At 400 m, the range in oxygen conditions was ~17 μmol kg$^{-1}$, similar to 300 m, but the CV

was higher (10.20%) because mean oxygen conditions were lower at this depth. High spatial resolution-sampling of the eastern tropical North Pacific OMZ documents considerable submesoscale oxygen variability with better oxygenated holes (Wishner et al. 2019); such patchiness could account for some of the variability we observed at ~400 m.

While we were only able to conduct one seasonal comparison, we observed that at 100 m, between the fall and spring deployments, mean oxygen conditions decreased from 132 to 104 μmol kg$^{-1}$ (Table 1), and there was little overlap in the oxygen measurements across the two deployments (Fig. 4, Fig. 7E). The most extreme high-oxygen conditions observed during the spring deployment (D100-DM-Spr) were equivalent to the most extreme low-oxygen conditions observed during the fall deployment (D100-DM-Fall) (Fig. 4).

CalCOFI data from nearby station 93.3 28 provides additional context on the characteristics of oxygen variability across seasonal and interannual timescales. When temperature and oxygen profiles from ~16 years of quarterly CalCOFI cruises are examined, we see that the highest temperature variability occurs in the upper water column (<50 m) and variability below ~150 m is relatively low (Fig. 7A,B). In contrast, absolute oxygen variability (i.e. standard deviation) is greatest between 50-150 m (Fig. 7C), and the coefficient of variation for oxygen (CV) actually increases below 100 m (Fig. 7D).

Comparing our high-frequency Nanolander deployment results to oxygen measurements across these ~16 years of quarterly CalCOFI cruises, we observe that the range in oxygen measurements at ~100 m, 300 m, and 400 m only captured a small portion of the variability measured across the ~16 year time period. In contrast, for the ~200 m deployments, a significant fraction of the variance over seasonal and interannual time-periods was captured by the short-term deployments (Fig. 7E-H). Oxygen variability in the SCB is also affected by the El Niño Southern Oscillation (ENSO), with oxygen conditions lower during La Niña periods (Nam et al. 2011). During the Nanolander deployments (August 2017-March 2018), the monthly Niño-3.4 index was always negative (-0.21 to -1.04; cpc.ncep.noaa.gov) but weaker than the La Niña conditions described in Nam et al. (2011). Our deployments, therefore captured a neutral ENSO/weak La Niña state. Interannual variability due to ENSO is captured in the data distribution from the CalCOFI cruises.

Across multidecadal scales, dissolved oxygen at ~100 m in the SCB dropped from 1984 to 2006 at a rate of 1.25-1.5 μmol kg$^{-1}$ year$^{-1}$ (Bograd et al. 2008). Over a period of ~20 years, this rate of oxygen loss equates to the seasonal difference at 100 m between the spring upwelling season and the fall. Thus, if this rate of oxygen loss continues, in 20 years, fall conditions would resemble current spring conditions. At 200 m, oxygen declines of 1-1.25 μmol kg$^{-1}$ year$^{-1}$ loss have been reported (Bograd et al. 2008), suggesting that if this same rate of oxygen decline continues, the mean oxygen conditions at these depths (which ranged from ~70-82 μmol kg$^{-1}$ from our data) will be continuously hypoxic in 10-20 years. Currently, communities are exposed to hypoxic conditions during the fall for <15% of the time in our time series (Table 1), but may experience hypoxic conditions more frequently in the spring. The greatest relative long-term changes in oxygen in the SCB have been reported at 300 m and represent an absolute change of 0.5-0.75 μmol kg$^{-1}$ year$^{-1}$ (Bograd et al. 2008). Conditions at 300 m were always hypoxic during our deployment, and may become more extreme in the future. At 400 m, oxygen

decreases of 0.25-0.5 µmol kg⁻¹ year⁻¹ have been reported (Bograd et al. 2008), and if these trends continue, in 13-26 years, this depth zone may become the upper boundary of the OMZ.

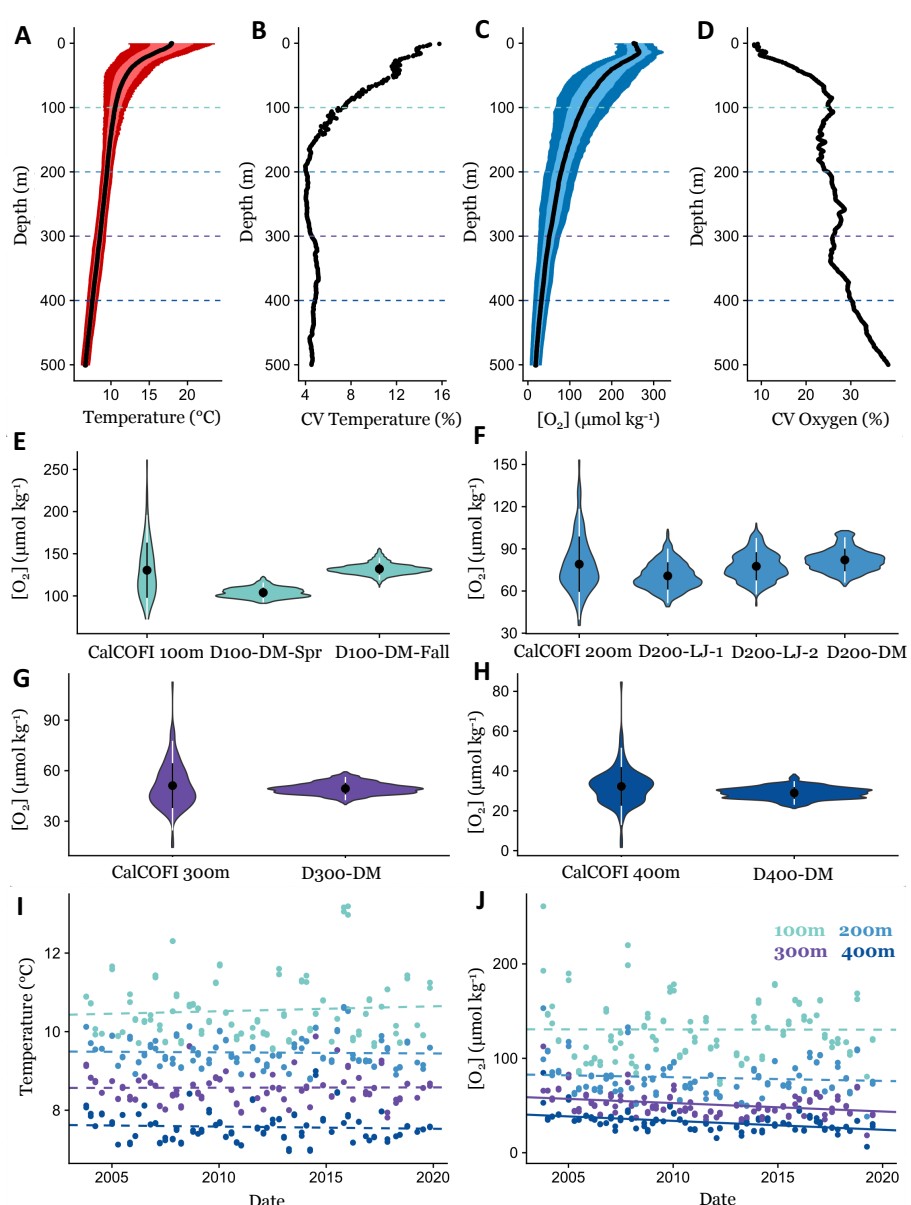

**Figure 7: Comparing short-term environmental variability from *DOV BEEBE* deployments to longer-term trends using CTD casts at nearby CalCOFI station (93.3 28.0).** Mean temperature (A) and oxygen (C) conditions through the water column (0-500 m) using CalCOFI CTD casts from Oct 2003-November 2019; light and dark colors indicate the variance around the mean and represent +/- 1 and 2 SD, respectively. Panels B and D show how the coefficient of variation (CV) for temperature and oxygen changes through the water column. Dotted lines in A-D indicate 100, 200, 300, and 400 m depths, and data are extracted for these depths for E-J. In E-H, violin plots show data distribution of oxygen measurements from ~16 years of CalCOFI quarterly cruises

 compared to ~3 week Nanolander deployments at 100 m (E), 200 m (F), 300 m (G), and 400 m (H). Violin plots show the mean +/- 1 SD (white) and +/- 2 SD (black). Panels I and J examine changes in temperature (I) and oxygen (J) conditions through time at 100, 200, 300, and 400 m. Dotted lines indicate non-significant linear relationships; solid lines indicate significant trends (p < 0.05).

While we have related our results to reported trends for the SCB from Bograd et al. (2008), it is unclear if these trends will continue, since multidecadal oxygen trends associated with the Pacific Decadal Oscillation (PDO) may reverse (McClatchie et al. 2010). In the 1950s and 1960s, oxygen levels were also very low in the SCB, and conditions at ~250 m were as low or lower than those reported in the early 2000s (McClatchie et al. 2010). Additionally, projections for the California Current System suggest winds near the equatorward boundary may weaken as winds strengthen in the northern region (Rykaczewski et al. 2016) leading to less coastal upwelling and higher oxygen conditions in the SCB.

In recent years (2003-2019), at the CalCOFI station closest to the Nanolander deployments (93.3 28), no significant linear deoxygenation trends were detected at 100 or 200 m, but significant deoxygenation trends were detected for 300 and 400 m (300 m: LR, $R^2$ = 0.10, p < 0.001; 400 m: LR, $R^2$ = 0.21, p < 0.001) (Fig. 7J). No significant warming trends were detected at these depths during this period (Fig. 7I). At 300 m, oxygen declined by 0.89 µmol kg$^{-1}$ year$^{-1}$ during the ~16 year time period, leading to a total oxygen loss of 14.25 µmol kg$^{-1}$ across the time series, and at 400 m oxygen declined by 0.94 µmol kg$^{-1}$ year$^{-1}$, leading to a total oxygen loss of 15.11 µmol kg$^{-1}$ over the ~16 years. Comparatively, the range of oxygen conditions experienced over the ~3-week Nanolander deployment was ~19 µmol kg$^{-1}$ at 300 m and ~17 µmol kg$^{-1}$ at 400 m.

## 4.2 Implications of environmental variability for seafloor communities

A recent FAO report on climate change impacts to deep sea fish and fisheries (FAO 2018) developed an index of exposure to climate hazard, which represents the mean changes in an environmental variable relative to its historical variability (defined by the standard deviation in the historic period). Higher historic variability reduces the index of exposure to climate hazard. Other studies also suggest that conditions of higher environmental variability may have a protective effect on the vulnerability of species to climate change. Frieder et al. (2014) concluded that high-frequency pH variability was an underappreciated source of pH-stress alleviation for invertebrates that were sensitive to low pH conditions. In a hypoxic fjord, slender sole, *Lyopsetta exilis*, were also observed to persist for short periods under mean oxygen conditions that were lower than their critical oxygen threshold ($P_{crit}$), which was likely facilitated by high oxygen variability around the mean and movements in and out of critically hypoxic waters (Chu et al. 2018). Many marine fish species show physiological plasticity in metabolic rate, gill surface area, and blood-oxygen binding curves in relation to short-term changes in oxygen conditions (Mandic et al. 2009, Nilsson 2010, Richards 2010, Dubruzzi and Bennett 2014).

Based on our results, benthic communities at ~200 m may be partially buffered from the negative effects of deoxygenation due to the substantial high-frequency variability of oxygen experienced over daily and weekly timescales. While at 400 m overall oxygen variability is lower than at 200 m, the amount of oxygen variability relative to the mean is similar to that at 200 m, suggesting variability may provide some reprieve to benthic communities at 400 m from low mean oxygen conditions and deoxygenation trends. At 400 m, conditions were severely hypoxic ($O_2$ < 22.5 µmol kg$^{-1}$) for ~1% of

the deployment time (Table 1), suggesting that even though this community is above the depth frequently associated with the upper boundary of the OMZ (450 m), it is already periodically exposed to OMZ conditions. Recent rapid deoxygenation trends at 400 m have also been observed at nearby CalCOFI station 93.3 28 from 2003-2019 (Fig. 7J).

Additionally, we note that between 200 and 300 m, there may be a boundary between two different water masses with implications for deoxygenation trends. The correlation between spiciness and oxygen concentration is negative at 200 m (indicative of high input of 13CW, which is a component of Pacific Equatorial Water, PEW), and then positive at 300 m (Fig. 5). Since changes in the volume of PEW have been implicated in the decreases in oxygen observed in the SCB (Booth et al. 2014, Bograd et al. 2015), it is worthwhile to note that increased input of this water mass could have a nonlinear effect on oxygen conditions in this area: increasing oxygen conditions at deeper depths, while decreasing them at shallower depths.

The high turbidity observed at 300 m may be due to shoaling and breaking nonlinear internal waves that can form bottom nepheloid layers (McPhee-Shaw 2006, Boegman and Stastna 2019). On the Peruvian margin, energy dissipation from tidally-driven internal waves have been shown to influence the distribution of epibenthic organisms by increasing suspension, transport, and deposition of food particles (Mosch et al. 2012). High turbidity conditions have also been observed during two separate ROV dives at ~340 m off Point Loma (unpublished, NDGallo), suggesting high turbidity conditions may be the norm at these depths on the upper slope in the SCB.

### 4.3 Observations of community responses to high-frequency environmental variability

Using *DOV BEEBE*, we were able to describe outer shelf and upper slope assemblages and examine if and how the seafloor community responds to $O_2$ variability at short timescales (daily, weekly, seasonal) in terms of community composition and diversity. Unexpectedly, we did not see strong evidence of seafloor community-level responses to daily and weekly oxygen variability. Seasonal differences were observed for D100-DM-Fall and D100-DM-Spr, but it is unclear if these were driven by oxygen, other upwelling-related environmental covariates such as temperature, pH, and productivity, or seasonal behavioral shifts associated with spawning or other activities.

Pronounced and rapid (< 2 week) community-level responses to hypoxia may occur in certain cases. For example, the Del Mar mooring has recorded strong event-based changes in dissolved oxygen (Nam et al. 2015) where oxygen rapidly increased or decreased over a short time-period (< 2 weeks). Rapid changes such as these that are outside of the typical regime of oxygen variability may lead to more immediate community responses. Unfortunately, we did not capture any such events during our deployments. Second, when oxygen conditions are near taxon-specific physiological thresholds, even small changes in oxygen can have large community-level effects (Levin et al. 2009, Wishner et al. 2018). In our deployments, spot prawn (*P. platyceros*) and crabs (*Cancer spp.*) were more strongly associated with the highest oxygen conditions during the D200-LJ-2 and D200-DM deployments, suggesting the oxygen conditions may be close to a critical threshold for these species. Tolerances to hypoxia are species-specific, with high intraspecies variability, so longer time series may better detect community-level changes. Given that we saw limited community-level responses at daily and

weekly timescales, future deployments could sample less frequently (one camera sample taken every hour or every two hours) but over longer time periods (~4-8 weeks) to examine community-level responses to environmental variability.

The lack of community-level response to diurnal and weekly oxygen variability seen in our data may not be surprising given that animals have several ways that they can respond to stressful conditions, which would not affect community-level abundance, diversity, or composition patterns. For example, fish can become less active and reduce metabolic demands (Richards 2009, 2010), or can decrease feeding behavior (Wu 2002, Nilsson 2010) during periodic hypoxia. We observed that animals were less active in the deeper deployments, but it is unclear if this is due to the lower oxygen conditions or other environmental covariates.

## 5.0 Concluding remarks

Ocean deoxygenation is a global concern, with changes in oxygen conditions potentially impairing the productivity of continental shelves and margins that support important ecosystem services and fisheries. Nanolanders provide a powerful tool to examine short-term, fine-scale fluctuations in nearshore dissolved oxygen and other environmental parameters, and associated ecological responses that are rarely recorded otherwise. Oxygen variability was strongly linked to tidal processes, and contrary to expectation, high-frequency oxygen variability did not decline linearly with depth. Depths of 200 and 400 m showed especially high oxygen variability and seafloor communities at these depths may be more resilient to deoxygenation stress because animals are exposed to periods of reprieve during higher $O_2$ conditions and may undergo physiological acclimation during periods of low $O_2$ conditions at daily and weekly timescales. Despite experiencing high oxygen variability, seafloor communities showed limited responses to changing conditions at these short timescales. However, our deployments did not capture any large acute changes in environmental conditions that may elicit stronger community responses; future studies using this platform could allow for such observations.

The Nanolander *DOV BEEBE* is configured to collect paired physical, biogeochemical, and biological data in the deep-sea over multiple days, which is a rarity except for in areas with developed ocean observatories. We found that *DOV BEEBE* performed well over the course of the deployments, and allowed us to study seafloor community responses to short-term environmental forcing. Our deployment lengths were limited by battery capacity to power the LED lights; all other elements would have allowed for longer sampling duration. Specific ways to extend future deployment lengths are currently being explored and include: using higher efficacy LEDs, integrating additional batteries to power the LED lights into newly devised Nanolander side pods, improving circuit performance that powers the LED lights by using new camera controllers and solid-state relays, and using low-light cameras, such as the Sony 7S II, which reduce the light required to illuminate the field of view. Longer deployment lengths would be advantageous for capturing ecosystem responses to environmental variability across time-scales (hours to months).

Many of the areas where large decreases in oxygen have been observed occur in developing countries, such as along the western and eastern coast of Africa (Schmidtko et al. 2017). Large oxygen losses have also been observed in the Arctic (Schmidtko et al. 2017), where the seafloor habitat is understudied. Due to their compact design, small landers such as *DOV*

*BEEBE* can provide a cost-effective and easily deployable tool for studying nearshore, deep-sea ecosystems and thus expand the capacity of developed and developing countries to monitor and study environmental changes along their coastlines. For continental margins and seafloor habitats, a global array of Nanolanders, similar in scope to the Argo program, could be
envisioned. These would provide coupled physical, biogeochemical, and ecological measurements, which would greatly expand our understanding of temporal and spatial heterogeneity in nearshore deep-sea ecosystems and seafloor community sensitivity to environmental change.

**Code and data availability**

Code and data are posted on Zenodo (http://doi.org/10.5281/zenodo.3897966). It is intended that elements of the Nanolander
design will be released as Open Source hardware on www.globaloceandesign.com within one year of publication.

**Author contribution**

NG, LL, KH, and NW designed the research plan; KH designed and built the Nanolander; NG, KH, and HY carried out the field deployments; NG, HY, and AN performed the video annotation and data analysis; NG wrote the manuscript and all authors contributed to editing the manuscript.

**Competing interests**

Author Kevin Hardy is the owner of the company Global Ocean Design which currently sells Nanolanders similar to *DOV BEEBE*.

**Acknowledgements**

This research was made possible through generous funding from several sources: the Mullin fellowship, Mildred E. Mathias
Research Grant, the Edna B. Sussman Fellowship, the Mia J. Tegner Fellowship, Friends of the International Center Scholarship, the *DEEPSEA CHALLENGE* Expedition, the National Science Foundation Graduate Research Fellowship, and the Switzer Environmental Leadership Fellowship. Additionally, Global Ocean Design LLC provided internal R&D resources. The Fisheries Society for the British Isles and the UCSD Graduate Student Association provided travel support to present these results at scientific meetings. This work would not have been possible without the support of a tremendous
number of people who helped with Nanolander deployments and recoveries, especially Phil Zerofski, Brett Pickering, Rich Walsh, Jack Butler, Mo Sedaret, Lilly McCormick, Andrew Mehring, Ana Sirovic, Rebecca Cohen, Ashleigh Palinkas, Jen McWhorter. I am forever grateful for the support of Javier Vivanco and others at Baja Aqua Farms for recovering and returning *DOV BEEBE* after it drifted into Mexican waters following an unsuccessful recovery. NG's Ph.D. committee

members, B. Semmens, R. Norris, R. Burton, D. Victor, and R. Keeling, provided feedback on the research. We thank Kevin Stierhoff, SungHyun Nam, and one additional anonymous reviewer for feedback on the manuscript. NG is currently supported by a NOAA QUEST grant to B. Semmens.

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
