# Peer review of "Characterizing deep-water oxygen variability and seafloor community responses using a novel autonomous lander"

_Biogeosciences, 2020_

## Referee Comment (RC1) · SungHyun Nam (Referee) · 25 Mar 2020

This paper presents novel results about oxygen variability over hours, days, and weeks and community composition at depths between 100 and 400 m along the nearshore environments in the Southern California Bight (SCB), reporting noble data collected using a new autonomous lander. I was very much interested to read the paper and sure that it will be of interest to many oceanographers; so I hope that this work can be published sooner than later. One of primary value of this paper lies in presenting high-frequency oxygen variability (periods of hours, days, and weeks) in different settings (depth, bathymetry, and season) of the SCB. I am impressed by the scope of the work

presented in this paper and by the envisioned global array of deep-sea landers that I had hoped to see for a decade. Authors presented a strong case convincing me (and readers to my mind) how deep-water oxygen variability can be characterized and that lower oxygen conditions to be associated with a transition from fish-dominated to invertebrate-dominated communities and that this taxonomic shift may be a useful ecological indicator of hypoxia. The hand-deployable Nanolander used in this study will be a powerful capacity-building tool for characterizing environmental variability and examining seafloor community sensitivity to climate-driven changes.

The manuscript is very well written, within the scope of Biogeosciences and informative, contributing to scientific progress substantially. No methodological flaws were detected. My overall recommendation is minor revision. The manuscript raises a few questions/comments some of which need to be addressed before publication to revise it in more attractable form. These are:

1) What is the bottom topography around the Nanolanders? Table 1 well summarizes the seven deployments including information on deployment location and depth. But, 'Scripps Coastal Reserve', 'Del Mar Steeples Reef' with latitudes/longitudes and bottom depth are not enough information for readers (particularly someone who is not familiar to the region) to figure out local bathymetric features, where outer shelf and upper slope are located/ranged/shaped, seafloor area exposed to different oxygen conditions, and so on. It is important to give details of the bathymetry around the deployment sites highlighting the key information as mentioned above. This would also be helpful for better discussing physical drivers of the oxygen variability. Thus, I would like to suggest to add one figure (or incorporated into Figure 1) showing compact and easy to understand map of the local bathymetry along with the deployment locations.

2) There is no summary/concluding remarks/conclusion in the manuscript. Substantial conclusions are reached but they are not presented as a separate section. Thus, I would like to suggest to add Section 5 to conclude or summarize the materials.

3) To give proper credit to related work, I would like to suggest to use '13CW', name of specific water mass linked to the deoxygenated water, instead of its locally defined water types, Pacific Equatorial Water (PEW) although previous works used the terms PEW. Based on recent work (Zachary et al., 2020; "The role of water masses in shaping the distribution of redox active compounds in the Eastern Tropical North Pacific oxygen deficient zone and influencing low oxygen concentrations in the eastern Pacific Ocean" published in Limnology and Oceanography as of 06 February 2020), two water masses – 13CW and deeper North Equatorial Pacific Intermediate Water (NEPIW) act as the two Pacific Equatorial source waters to the California Current System corresponding to upper and lower PEW at isopycnals of 26.2-26.8 kg m-3 when defined locally. Here, the relevant water mass seems to be 13CW (upper PEW), and not NEPIW (lower PEW).

4) The observed oxygen variability over short time scales was compared with multi-decade-long deoxygenation or long-term trends/shifts reported in Bograd et al. (2008) and McClatchie et al. (2010). However, it was not discussed in comparison to inter-annual oxygen variability in the region. Does the period of data collection from August 2017 to March 2018 correspond to normal or more likely abnormal (El Niño/La Niña) year? My suggestion is to provide discussions on the observational results in terms of significant local interannual oxygen variability in association with such large-scale condition presented in Nam et al. (2011; "Amplification of hypoxia and acidic events by La Niña conditions on the continental shelf off California" published in Geophysical Research Letters as of 23 November 2011).

5) What are depths of thermocline/oxycline (any strong vertical temperature/oxygen gradient close to 200 m?) and their sectional structures across the shelf-slope? It would be helpful to check the cross-sectional structures of water temperature and dis-solved oxygen across the shelf and slope at a given time, e.g., see Figure 2 of Nam et al. (2011) but focusing on the deeper area (over the slope). Both mean and standard deviation to the mean, thus the CV of the temperature/oxygen can be partly explained from its vertical (and horizontal) gradient. My question is whether relatively high CV is

due to strong vertical (or horizontal) gradient of temperature and oxygen (thermocline and oxycline depths). Also, how the structures are different from spring (D100-DM-Spr) vs. fall (DM100-DM-Fall)? It would also be relevant to high turbidity condition around 300 m as the internal waves/internal tides break and enhance the mixing (to resuspend the sediment) when and where the isopycnals (isotherms) touch the bottom (see the comments #6 below for details).

6) As described in Abstract, the high-frequency oxygen variability was strongly linked to tidal processes. But, I do not understand why it is contrary to expectation. As described in Section 1 (Lines 54-57), Section 3 (Lines 308-313), Section 4 (Lines 449-450 and 479-480), and Supplements, diurnal and semidiurnal oxygen variability is noticeable. This is not something unexpected but consistent with previous works reporting oxygen variability in a shallower zone, e.g., Frieder et al. (2012). Importance of tidal processes may also be confirmed from spring-neap cycles or modulations of semidiurnal/diurnal oxygen fluctuations. I could see such a spring (neap) amplification (reduction), for example, from time series plot of D10 - 98 m or D100-DM-Spr in Supplement 1B. Amplitudes of semidiurnal oxygen fluctuations reach up to larger than 20 $\mu$mol kg-1 for Days 0-3 and 10-13 (presumably corresponding to spring tide) while smaller than 10 $\mu$mol kg-1 for Days 5-8 and 17-20 (presumably corresponding to neap tide). What are CVs for periods of spring vs neap tides?

I believed and continue to believe that such high-frequency oxygen variability is relevant to internal tides generated and shoaled at a specific phase of the surface tide in a sloping bottom (even up to the zone as shallow as 15 m) as reported in the region by Nam and Send (2011) and others. It is generally known that the isotherms (so iso-oxygen surfaces) move up and down at high-frequency due to propagation and evolution internal tides and associated shorter period nonlinear internal waves (also termed internal solitary waves). When they shoal and break, turbulent mixing is markedly enhanced often forming bottom nepheloid layer that may account for suspended sediments and the high turbidity condition around 300 m. The bottom nepheloid layer has been presented since McPhee-Shaw (2006; "Boundary-interior exchange: Reviewing the idea that internal-wave mixing enhances lateral dispersal near continental margins" published in Deep Sea Research II: Topical studies in Oceanography as of 20 February 2006), e.g., Boegman and Stastna (2019; "Sediment resuspension and transport by internal solitary waves" published in Annual Reviews of Fluid Mechanics as of 15 August 2018).

7) Not being a biologist, I do not know in detail how the seafloor communities respond to short-period (mostly diurnal) changes in environmental conditions, but it is convincing that longer time series data are vital for addressing the science issue. My question is why camera sample should be less frequent for longer-term deployment. Is it limited by battery or memory? There would be several technical ways to overcome battery or memory limit. Why not trying new technologies that allow longer-term deployment keeping the same camera (as well as other sensors) sampling frequency.

---

## Referee Comment (RC2) · Anonymous Referee #2 · 7 Apr 2020

General comments: This paper addresses the pressing problem of progressing ocean deoxygenation and the effect of variable oxygen availability on fish and epibenthic invertebrate communities along a depth gradient of 100 to 400m off San Diego, CA. For time series measurements of oxygen and other environmental parameters, the authors use a novel lander system, which harbors a SBE CTD, oxygen sensors and a camera/light system. The camera provides video sequences, which were used for the community analysis. The introduction of this novel lander system is a major focus of the MS. This study represents an interesting approach of how benthic community data can be obtained and related to physico-chemical time series measurements, it is based on an extended data set of seven lander deployments and might be very interesting for

a wider marine ecological community of scientists.

The paper is very well and concisely written. The manuscript is well structured, the methods are appropriate and the data, which are novel are well presented. The authors being well aware that their study does (or better say can only include) a limited number of environmental parameters are careful about their conclusions and present those in an appropriate way.

The MS has a few very minor weaknesses that can be easily solved, which I would like to raise in the following.

I very much like the idea of the small-sized and hand-operated lander and I fully agree with the need for such systems for the performance of more in situ long-term observations. Yet, given the actual size and weight of the lander, the expression 'Nanolander' seems a bit exaggerated. This is just a personal opinion and is by no means meant to urge the authors to change the name of their system. In this context, the last section of the MS "A global array of deep-sea landers" goes in the same direction and appears a bit superficial with an emphasis on "selling" the system. The authors might consider to rewrite this last section increasing its profoundness.

As the paper claims to introduce a novel lander-technology, I would have wished to find a brief review of similar already existing systems. The authors mention papers by Jamieson et al. but do not provide details. Please add a few lines highlighting where your system goes beyond existing systems.

Beside oxygen, other parameters were measured (temperature, pH, saturation state of aragonite/calcite) but these were hardly mentioned in the discussion section although e.g. pH in respiration physiology is very important. Please clarify why these parameters were not further included in the interpretation of the data set.

Further comments and edits:

Line 24: please explain "phest"

Line 67-72 in this context eddy correlation techniques could be mentioned

Line 108: suggest to use only metric units of m or cm instead of ft

Figure 1, suggest to include a more detailed technical drawing of the lander (i.e. better version of Fig. 1A) where the different major components are labeled with numbers which can referred to in the main text. Figure 1D is not really providing any additional information and could be omitted. Please provide in the final version of the MS the figures in sufficient resolution.

Line 111: "glass filled" sounds a bit odd; do you mean glass-spheres housed by polyamide protective shells

Line 122: "The power supply for the BART board is housed in the upper sphere", together with the Bart board?

Line 123ff: what would be the maximum deployment time of DOV Beebe with the given battery systems?

Line 131: would be nice if especially details of the camera system could better show up in the improved version of Figure 1A

Line 153: please use metric units

Line 178: I think there is no need to use the word "high-frequency" (it's rather a matter of the perspective whether 5 min sampling rate is high-frequency or not)

Line 194 please describe spiciness in a bit more detail, it's likely not common to everybody

Line 309 deconstructed time series - please explain in more detail

Figure 4: the labels for "day" and "night" are difficult to read – please enlarge

Line 448: I am not sure whether the statement "At ∼200 m, oxygen, temperature, and pH exhibited high variability (Fig. 2), greater at times than the variability observed

at 100 m." is correct for temperature – please check. Although the Figure 2 is quite attractive and informative, especially for the discussion section, when environmental variability is discussed additional Box plots might be helpful to elucidate the differences between the different deployments (i.e. depths).

Line 476 Turbidity can be related to local hydrodynamics caused by the energy dissipation of incipient internal tides at sloping boundaries affecting the suspension, transport and deposition of food particles. If you are interested, please see e.g. Mosch et al. (2012) Factors influencing the distribution of epibenthic megafauna across the Peruvian oxygen minimum zone. Deep-Sea Research I 68 (2012) 123–135 and references therein.

---

## Short Comment (SC1) · 24 Apr 2020

Hi

I wonder if the authors might provide a list of the fish species they documented with the minimum and maximum depths these taxa were encountered?

---

## Author Comment (AC1) · 23 May 2020

We are very grateful to the reviewer for his thorough review of our manuscript, and for the concrete suggestions on how to further improve this study. We respond to each comment below.

1) What is the bottom topography around the Nanolanders? Table 1 well summarizes the seven deployments including information on deployment location and depth. But, 'Scripps Coastal Reserve', 'Del Mar Steeples Reef' with latitudes/longitudes and bottom depth are not enough information for readers (particularly someone who is not familiar to the region) to figure out local bathymetric features, where outer shelf

[Figure]

and upper slope are located/ranged/shaped, seafloor area exposed to different oxygen conditions, and so on. It is important to give details of the bathymetry around the deployment sites highlighting the key information as mentioned above. This would also be helpful for better discussing physical drivers of the oxygen variability. Thus, I would like to suggest to add one figure (or incorporated into Figure 1) showing compact and easy to understand map of the local bathymetry along with the deployment locations.

RESPONSE: Thank you for highlighting this omission. We have created a new figure (called New Figure 3) which clearly shows the deployment locations (green diamonds) in relationship to local bathymetric features. Despite relatively close spatial proximity between the Scripps Reserve and Del Mar deployment sites (∼10 km), there are important bathymetric differences. The Scripps Coastal Reserve deployment sites are positioned close to a submarine canyon feature (the La Jolla canyon), while the Del Mar Steeples Reef deployment sites are on a gradually sloping margin. Additionally, we have added the locations of nearby CalCOFI stations (93.3 26.7 and 93.3 30) (black circles) and have included data from CalCOFI station 93.3 30 to provide additional context regarding variability over longer timescales.

2) There is no summary/concluding remarks/conclusion in the manuscript. Substantial conclusions are reached but they are not presented as a separate section. Thus, I would like to suggest to add Section 5 to conclude or summarize the materials.

RESPONSE: Thank you for this suggestion. We have modified what was previously section 4.3 of the discussion, titled "A global array of deep sea landers", added additional content summarizing the findings of the study, and titled this section: "Section 5: Concluding remarks". This section now reads:

"5.0 Concluding remarks

Ocean deoxygenation is a global concern, with changes in oxygen conditions potentially impairing the productivity of continental shelves and margins that support important ecosystem services and fisheries. Nanolanders provide a powerful tool to examine short-term, fine-scale fluctuations in nearshore dissolved oxygen and other environmental parameters, and associated ecological responses that are rarely recorded otherwise. Oxygen variability was strongly linked to tidal processes, and contrary to expectation, oxygen variability did not decline linearly with depth. Depths of 200 and 400 m showed especially high oxygen variability which may buffer communities at these depths to deoxygenation stress by exposing them to periods of relatively high oxygen conditions across short timescales (daily and weekly). Despite experiencing high oxygen variability, seafloor communities showed limited responses to changing conditions at these short time-scales. However, our deployments did not capture any large acute changes in environmental conditions, that may elicit stronger community responses; future studies using this platform could allow for such observations.

Nanolanders provide a cost-effective and easily deployable tool for studying local conditions throughout the world. Many of the areas where large decreases in oxygen have been observed occur in developing countries, such as along the western and eastern coast of Africa (Schmidtko et al. 2017). Large oxygen losses have also been observed in the Arctic (Schmidtko et al. 2017), where the seafloor habitat is understudied. Due to their compact design, small landers such as DOV BEEBE can provide easy access to nearshore, deep-sea ecosystems and could expand the capacity of developed and developing countries to monitor and study environmental changes along their coastlines. We found that the Nanolander performed well and reliably over the course of the deployments, and allowed us to study seafloor community responses within the context of short-term environmental forcing. For continental margins and seafloor habitats, a global array of Nanolanders, similar in scope to the Argo program, could be envisioned. These would provide coupled physical, biogeochemical, and ecological measurements, which would greatly expand our understanding of temporal and spatial heterogeneity in nearshore deep-sea ecosystems and seafloor community sensitivity to environmental change. "

3) To give proper credit to related work, I would like to suggest to use '13CW', name

of specific water mass linked to the deoxygenated water, instead of its locally defined water types, Pacific Equatorial Water (PEW) although previous works used the terms PEW. Based on recent work (Zachary et al., 2020; "The role of water masses in shaping the distribution of redox active compounds in the Eastern Tropical North Pacific oxygen deficient zone and influencing low oxygen concentrations in the eastern Pacific Ocean" published in Limnology and Oceanography as of 06 February 2020), two water masses – 13CW and deeper North Equatorial Pacific Intermediate Water (NEPIW) act as the two Pacific Equatorial source waters to the California Current System corresponding to upper and lower PEW at isopycnals of 26.2-26.8 kg m-3 when defined locally. Here, the relevant water mass seems to be 13CW (upper PEW), and not NEPIW (lower PEW).

RESPONSE: Thank you for drawing our attention to this new reference. We have added the following clarification in the manuscript. In Section 2.2: "Previous studies have found that changes in oxygen and pH in the Southern California Bight are associated with changes in the volume of Pacific Equatorial Water (PEW) transported in the California Undercurrent (Bograd et al. 2015, Nam et al. 2015). PEW is characterized by low oxygen, warm, high salinity conditions, and is composed of two watermasses, the 13°C water mass (13CW) and the deeper Northern Equatorial Pacific Intermediate Water Mass (NEPIW) (Evans et al. 2020)."

Further, in the discussion, we have updated our reference to PEW to 13CW and included the appropriate citation: "Input of 13°C water (13CW), which is brought up by the California Undercurrent (Evans et al. 2020), is key to determining near-seafloor oxygen conditions at ∼200 m, and in the spring, 13CW upwells to 100 m leading to lower oxygen conditions. In contrast, at deeper depths (∼300 and 400 m), added input of 13CW increases oxygen conditions."

In Section 4.1, we have also updated the text to read, "We note that 300 m is an interesting depth which may be at an important boundary between two different water masses. The correlation between spiciness and oxygen concentration is negative at 200 m (indicative of high input of 13CW, which is a component of Pacific Equatorial

Water), and then positive at 300 m (Fig. 3)."

To maintain consistency with the nomenclature in the reference, we have kept the use of "PEW" in cases where it directly refers to the results of a study. For example, "Previous studies have found that changes in oxygen and pH in the Southern California Bight are associated with changes in the volume of Pacific Equatorial Water (PEW) transported in the California Undercurrent (Bograd et al. 2015, Nam et al. 2015)."

4) The observed oxygen variability over short time scales was compared with multi-decade-long deoxygenation or long-term trends/shifts reported in Bograd et al. (2008) and McClatchie et al. (2010). However, it was not discussed in comparison to inter-annual oxygen variability in the region. Does the period of data collection from August 2017 to March 2018 correspond to normal or more likely abnormal (El Niño/La Niña) year? My suggestion is to provide discussions on the observational results in terms of significant local interannual oxygen variability in association with such large-scale condition presented in Nam et al. (2011; "Amplification of hypoxia and acidic events by La Niña conditions on the continental shelf off California" published in Geophysical Research Letters as of 23 November 2011).

RESPONSE: Thank you for this suggestion. Indeed, we were interested in comparing how short-term variability compares to longer-term variability driven both by interannual and multidecadal changes as one of the objectives of this research. Between August 2017 and March 2018, the conditions in the Eastern Pacific were more consistent with La Niña conditions; associated with being lower in oxygen and pH on average (as shown in Nam et al. 2011). The monthly Niño-3.4 index was always negative during the period of our data collection and ranged from -0.21 to -1.04 (https://www.cpc.ncep.noaa.gov/products/analysis_monitoring/ensostuff/detrend.nino34.ascii.txt). However, these conditions were much weaker than the La Niña time-period (Jul 2010-Jan 2011) described in Nam et al. (2011) during which the monthly Niño-3.4 index ranged between -1.04 to -1.73.

[Figure]

To compare our high-frequency measurements to a longer-term dataset, we incorporated data from a nearby CalCOFI station to provide additional context to our results. We relied on data from CalCOFI Station 93.3 28 since it was the closest station to our deployments which sampled the full upper water column down to 500 m. CalCOFI Station 93.3 26.7 was too shallow for our comparison; but both stations are provided in the new map figure with deployment sites (New Figure 3). We then used all available CTD casts for Station 93.3 28, which represented data from 65 CalCOFI cruises during the time-period between July 2003 and November 2019, and looked at how the overall variability in environmental conditions across this longer (∼16 year) time-period, compares to the overall variability in environmental conditions across our shorter (∼3-week deployments). These results are presented in a new figure labeled New Figure 7. This figure shows how the mean, variance (indicated using +/- 1SD and +/- 2SD), and coefficient of variation (CV) for temperature and oxygen change across the upper 500m of the water column at Station 93.3 28 (Panels A-D). This figure also selects data from specific depths that relate to our targeted deployment depths (100, 200, 300, and 400 m), and shows how the variance distribution in temperature and oxygen across our ∼3 week deployments compares to the observed variance at these depths over ∼16 years of CalCOFI cruise measurements (Panels E-H). Additionally, we have looked for evidence of linear changes in temperature or oxygen at our targeted deployment depths (100, 200, 300, and 400 m) at CalCOFI Station 93.3 28 (Panels I-J) as additional context for longer-term change. We hope that this added analysis helps frame our results regarding variability over short timescales within the context of variability over interannual and multidecadal timescales.

5) What are depths of thermocline/oxycline (any strong vertical temperature/oxygen gradient close to 200 m?) and their sectional structures across the shelf-slope? It would be helpful to check the cross-sectional structures of water temperature and dissolved oxygen across the shelf and slope at a given time, e.g., see Figure 2 of Nam et al. (2011) but focusing on the deeper area (over the slope). Both mean and standard deviation to the mean, thus the CV of the temperature/oxygen can be partly explained

from its vertical (and horizontal) gradient. My question is whether relatively high CV is due to strong vertical (or horizontal) gradient of temperature and oxygen (thermocline and oxycline depths). Also, how the structures are different from spring (D100-DM-Spr) vs. fall (DM100-DM-Fall)? It would also be relevant to high turbidity condition around 300 m as the internal waves/internal tides break and enhance the mixing (to resuspend the sediment) when and where the isopycnals (isotherms) touch the bottom (see the comments #6 below for details).

RESPONSE: To look at patterns in cross-sectional structures of water temperature and dissolved oxygen across the shelf and slope, and to look at how these spatial patterns change seasonally, we extracted data from CalCOFI stations 93.3 28, 93.3 30, 93.3 35, and 93.3 40 and examined the CTD profiles for these stations during the deployment period. Four cruises were relevant to examine, however, cruise 1802SH was shortened due to the government shutdown and therefore only one of the four stations (93.3 30) was sampled. As such, we focused this additional analysis on just the three cruises (1708SR – August 2017, 1711SR – November 2017, and 1804SH – April 2018). In the new supplementary figure (attached, and titled New Supplementary 1), we show the temperature and oxygen profiles for these four stations across the three relevant cruises. From these profiles, we see that in the spring (April 2018), there is no onshore-offshore gradient, whereas in summer (August 2017) and to a lesser degree in late fall (November 2017), spatial differences in onshore (93.3 28 and 93.3 30) and offshore (93.3 35 and 93.3 40) environmental profiles are evident. These spatial differences are most pronounced in late summer (August 2017). Additionally, in August 2017 there is evidence of some unusual vertical structure in the oxygen profile around ∼200 m; both at station 93.3 28 and 93.3 30. Our first deployment (D200-LJ-1) was conducted in late August, so may have captured part of this feature. However, this cannot fully explain the higher variability we observed at 200 m, because our later deployment (D200-DM) was done in mid November, when there is no evidence of unusual vertical structure in the oxygen profile at 200 m for 93.3 28 or 93.3 30. These supplementary profiles, as well as the profiles in New Figure 7 do show that the thermocline is steeper and

shallower, overall, than the oxycline. We hope these additional datapoints help shed light on the sources of observed variability in our short-term deployments.

6) As described in Abstract, the high-frequency oxygen variability was strongly linked to tidal processes. But, I do not understand why it is contrary to expectation. As described in Section 1 (Lines 54-57), Section 3 (Lines 308-313), Section 4 (Lines 449-450 and 479-480), and Supplements, diurnal and semidiurnal oxygen variability is noticeable. This is not something unexpected but consistent with previous works reporting oxygen variability in a shallower zone, e.g., Frieder et al. (2012). Importance of tidal processes may also be confirmed from spring-neap cycles or modulations of semidiurnal/diurnal oxygen fluctuations. I could see such a spring (neap) amplification (reduction), for example, from time series plot of D10 - 98 m or D100-DM-Spr in Supplement 1B. Amplitudes of semidiurnal oxygen fluctuations reach up to larger than 20 $\mu$mol kg-1 for Days 0-3 and 10-13 (presumably corresponding to spring tide) while smaller than 10 $\mu$mol kg-1 for Days 5-8 and 17-20 (presumably corresponding to neap tide). What are CVs for periods of spring vs neap tides? I believed and continue to believe that such high-frequency oxygen variability is relevant to internal tides generated and shoaled at a specific phase of the surface tide in a sloping bottom (even up to the zone as shallow as 15 m) as reported in the region by Nam and Send (2011) and others. It is generally known that the isotherms (so iso-oxygen surfaces) move up and down at high-frequency due to propagation and evolution internal tides and associated shorter period nonlinear internal waves (also termed internal solitary waves). When they shoal and break, turbulent mixing is markedly enhanced often forming bottom nepheloid layer that may account for suspended sediments and the high turbidity condition around 300 m. The bottom nepheloid layer has been presented since McPhee-Shaw (2006; "Boundary-interior exchange: Reviewing the idea that internal-wave mixing enhances lateral dispersal near continental margins" published in Deep Sea Research II: Topical studies in Oceanography as of 20 February 2006), e.g., Boegman and Stastna (2019; "Sediment resuspension and transport by internal solitary waves" published in Annual Reviews of Fluid Mechanics as of 15 August 2018).

RESPONSE: Thank you for raising these points. One of the objectives of this study was to place rates of anthropogenic change within the context of short-term variability that nearshore deep-sea communities are exposed to. These results show that tidally-driven variability is an important source of high-frequency variability to consider, that could either exacerbate or buffer deep-sea communities from changes in mean conditions with climate change. Contrary to the idea that the deep-sea is a stable environment, these results show a substantial amount of environmental variability occurring at short timescales on the upper margin.

As suggested by the reviewer, we examined the CVs for the two spring and neap periods captured during D100-DM-Spr. The CVs for the two time periods corresponding to the spring tide (Days 0-3 and Days 10-13) were 5.02% and 4.76%, respectively, while the CVs for the two time periods corresponding to neap tide (Days 5-8 and 17-20) were 2.66% and 2.69%, respectively.

Additionally, we have added the following information in the discussion in section 4.1: "The high turbidity observed at this depth may be due to shoaling and breaking nonlinear internal waves that can form bottom nepheloid layers (McPhee-Shaw 2006, Boegman and Stastna 2019). High turbidity conditions have also been observed during two separate ROV dives at ~340 m off Point Loma (unpublished, NDGallo), suggesting high turbidity conditions may be the norm at these depths on the upper slope in the SCB."

7) Not being a biologist, I do not know in detail how the seafloor communities respond to short-period (mostly diurnal) changes in environmental conditions, but it is convincing that longer time series data are vital for addressing the science issue. My question is why camera sample should be less frequent for longer-term deployment. Is it limited by battery or memory? There would be several technical ways to overcome battery or memory limit. Why not trying new technologies that allow longer-term deployment keeping the same camera (as well as other sensors) sampling frequency.

RESPONSE: Ideally, we would like to maintain both the high-resolution sampling frequency (20 second video samples every 20 minutes) and extend the deployment length. In this study, the main technological limitation we ran into was limited battery capacity to power the LED lights; all other elements would have allowed for longer sampling (camera battery and memory, SBE MicroCAT battery and memory). The basic Nanolander itself can stay in situ for periods of 2 years, perhaps longer. However, we were only able to provide sufficient power to the LED lights for a maximum period of 14 days at the selected sampling frequency.

As we see the advantages of even longer time series, we are looking at the limitations of our initial technology choices. Some of these were made on the basis of cost and availability, others because of familiarity. We have looked specifically at ways to improve the power capacity to the LED lights, which can be done by using new camera controllers, solid-state relays, and high efficacy LEDs. Additional batteries to power the LED lights can also be added by integrating them into newly devised side pods that attach to the Nanolander. Using low light cameras, such as the Sony $\alpha$7S II, would also reduce the amount of LED light required, reducing power drain, and would increase the depth of field. These options are all currently being explored as ways to extend future deployment lengths, while maintaining the high-resolution sampling frequency.
* * *
**Fig. 1.** New Figure 3

[Figure]

**Fig. 2.** New Figure 7

**A** August 2017 (1708SR)
**B** November 2017 (1711SR)
**C** April 2018 (1804SH)

Depth (m)
Temperature (°C)

CalCOFI Station
- 093.3 028.0
- 093.3 030.0
- 093.3 035.0
- 093.3 040.0

**D**
**E**
**F**

Depth (m)
$[O_2]$ (µmol kg$^{-1}$)

CalCOFI Station
- 093.3 028.0
- 093.3 030.0
- 093.3 035.0
- 093.3 040.0

**Fig. 3.** New Supplementary 1

---

## Author Comment (AC2) · 23 May 2020

We are happy to provide a table with the list of fish species observed, including minimum and maximum depths these taxa were encountered. Additionally, we have included the temperature and oxygen conditions for these observations. We propose to include this Table as a Supplement to the manuscript.

[Figure]

Interactive
comment

| Fish Species | Deployment | Depth Range (m) | Temperature (°C) | Oxygen (μmol/kg) | Number of Video Clips Observed in |
|---|---|---|---|---|---|
| *Anarrhichthys ocellatus* | D100-DM-Fall | 99 | 11.25 | 134.40 | 1 |
| *Apristurus brunneus* | D400-DM | 399 | 7.27- 7.50 | 26.70 - 30.76 | 4 |
| *Cephaloscyllium ventriosum* | D200-LJ-2 | 178 | 9.78 - 10.01 | 73.38 - 89.47 | 4 |
| *Chilara taylori* | D200-LJ-2, D100-DM-Fall, D100-DM-Spr | 98 - 178 | 9.42 - 11.56 | 49.51 - 142.43 | 358 |
| *Eptatretus stoutii* | D200-LJ-2, D100-DM-Fall, D200-DM, D400-DM, D100-DM-Spr | 98 - 399 | 7.10 - 11.46 | 22.54 - 138.67 | 124 |
| *Faciolella equatorialis* | D200-LJ-2, D400-DM | 399 - 178 | 7.07 - 10.00 | 21.55 - 90.32 | 130 |
| *Genyonemus lineatus* | D200-LJ-2 | 178 | 9.60 - 10.12 | 62.04 - 91.47 | 11 |
| *Hydrolagus colliei* | D200-LJ-2, D100-DM-Fall, D200-DM, D400-DM, D100-DM-Spr | 98 - 399 | 7.26 - 11.46 | 24.76 - 138.67 | 79 |
| *Lycodes sp.* | D200-LJ-2, D100-DM-Fall, D200-DM, D100-DM-Spr | 98 - 192 | 9.19 - 11.60 | 50.18 - 144.76 | 283 |
| *Lyconema barbatum* | D200-LJ-2, D100-DM-Spr | 98 - 178 | 9.86 - 10.09 | 74.44 - 109.69 | 4 |
| *Lyopsetta exilis* | D400-DM | 399 | 7.18 | 23.02 | 1 |
| *Merluccius productus* | D200-DM, D300-DM, D400-DM, D100-DM-Spr | 98 - 399 | 7.47 - 9.86 | 29.90 - 108.68 | 11 |
| *Microstomus pacificus* | D200-LJ-2, D400-DM | 178 - 399 | 7.10 - 10.15 | 22.54 - 96.72 | 100 |
| *Nezumia liolepis* | D400-DM | 399 | 7.19 - 7.39 | 23.43 - 29.63 | 9 |
| *Ophiodon elongatus* | D100-DM-Fall, D100-DM-Spr | 98 - 99 | 9.63 - 11.29 | 103.2 - 134.3 | 9 |
| *Osteichthyes unknown* | D200-LJ-2, D100-DM-Fall, D200-DM, D100-DM-Spr | 98 - 192 | 9.19 - 11.89 | 49.51 - 146.99 | 414 |
| *Oxylebius pictus* | D100-DM-Fall | 99 | 11.26 | 136.90 | 1 |
| Pacific sanddab | D200-LJ-2, D200-DM | 178 - 192 | 9.25 - 9.79 | 64.30 - 91.22 | 21 |
| *Physiculus rastrelliger* | D200-LJ-2, D100-DM-Fall, D100-DM-Spr | 98 - 178 | 9.59 - 11.27 | 65.33 - 133.62 | 12 |
| *Raja rhina* | D400-DM | 399 | 7.15 - 7.44 | 23.57 - 30.20 | 2 |
| *Rathbunella hypoplecta* | D200-LJ-2, D100-DM-Fall, D100-DM-Spr | 98 - 178 | 9.50 - 11.89 | 62.04 - 146.99 | 318 |
| *Scorpaena gutatta* | D100-DM-Fall | 99 | 11.03 | 132.26 | 1 |
| *Sebastes caurinus* | D100-DM-Fall | 99 | 10.48 - 11.20 | 119.2 - 133.10 | 3 |
| *Sebastes diploproa* | D400-DM | 399 | 7.32 | 27.11 | 1 |
| *Sebastes elongatus* | D100-DM-Fall, D100-DM-Spr | 98 - 99 | 9.56 - 11.59 | 100.27 - 142.16 | 40 |
| *Sebastes lentiginosus* | D100-DM-Fall | 99 | 10.80 - 11.60 | 126.60 - 144.80 | 13 |
| *Sebastes miniatus* | D100-DM-Fall | 99 | 10.63 - 11.24 | 124.0 - 137.3 | 15 |
| *Sebastes ovalis* | D100-DM-Fall | 99 | 11.17 - 11.50 | 131.1 - 139.8 | 3 |
| *Sebastes paucispinis* | D100-DM-Fall, D100-DM-Spr | 98 -99 | 9.92 - 11.24 | 109.60 - 137.20 | 27 |
| *Sebastes pinniger* | D100-DM-Fall, D100-DM-Spr | 98 -99 | 9.52 - 11.89 | 99.15 - 146.99 | 78 |
| *Sebastes rosaceus* | D100-DM-Fall, D100-DM-Spr | 98 -99 | 9.64 - 11.55 | 106.65 - 141.20 | 38 |
| *Sebastes rubrivinctus* | D100-DM-Fall, D100-DM-Spr | 98 -99 | 9.66 - 11.46 | 104.3 - 138.7 | 60 |
| *Sebastes semicinctus* | D200-LJ-2, D100-DM-Fall, D100-DM-Spr | 98 - 178 | 9.42 - 11.89 | 74.42 - 146.99 | 1227 |
| *Sebastes Sp  Dark orange* | D100-DM-Spr | 98 | 9.69 - 10.21 | 106.3 - 122.7 | 2 |
| *Sebastes sp  Unidentified* | D200-LJ-2, D100-DM-Fall, D200-DM, D100-DM-Spr | 98 -192 | 9.03 - 11.79 | 58.69 - 146.22 | 626 |
| *Sebastolobus alascanus* | D400-DM | 399 | 7.19 - 7.49 | 23.41 - 30.61 | 38 |
| *Seriphus politus* | D100-DM-Fall | 99 | 10.92 | 128.90 | 1 |
| Skate | D200-LJ-2 | 178 | 9.86 | 76.89 | 1 |
| *Squalus suckleyi* | D200-DM | 192 | 9.41 | 71.73 | 1 |
| *Symphurus sp* | D200-LJ-2 | 178 | 9.76 - 10.29 | 68.87 - 97.94 | 15 |
| *Synodus lucioceps* | D200-LJ-2, D100-DM-Fall, D200-DM | 99 - 192 | 9.22 - 11.50 | 56.80 - 139.78 | 210 |
| Unidentified shark | D100-DM-Fall, D200-DM | 99 - 192 | 9.57 - 10.71 | 71.49 - 124.82 | 2 |
| *Xeneretmus latifrons* | D200-LJ-2, D200-DM, D400-DM | 178 - 399 | 7.08 - 10.28 | 22.27 - 95.11 | 324 |
| *Zalembius roseaceus* | D100-DM-Fall, D100-DM-Spr | 98 - 99 | 9.90 - 11.34 | 109.5 - 140.2 | 13 |
| *Zaniolepis frenata* | D100-DM-Fall | 99 | 10.80 - 11.31 | 126.6 - 137.3 | 5 |
| *Zaniolepis spp* | D200-LJ-2, D100-DM-Fall, D200-DM, D100-DM-Spr | 98 - 192 | 9.12 - 11.73 | 64.86 - 146.99 | 220 |

**Fig. 1.** Fish Observations Table

---

## Author Comment (AC3) · 23 May 2020

I very much like the idea of the small-sized and hand-operated lander and I fully agree with the need for such systems for the performance of more in situ long-term observations. Yet, given the actual size and weight of the lander, the expression 'Nanolander' seems a bit exaggerated. This is just a personal opinion and is by no means meant to urge the authors to change the name of their system. In this context, the last section of the MS "A global array of deep-sea landers" goes in the same direction and appears a bit superficial with an emphasis on "selling" the system. The authors might consider to rewrite this last section increasing its profoundness.

[Figure]

RESPONSE: We are very grateful to the reviewer for their thoughtful and thorough review of our manuscript and for their additional suggestions for improvement. We respond to each comment in bold.

We agree in principle that the small benthic lander is not actually "nano", however, Nanolander is now a registered proper name that specifies this scientific platform, so we maintain the name in the manuscript. The term arose to distinguish it from much larger traditional benthic landers.

We appreciate the reviewer's perspective that the final section could be further expanded to increase its profoundness. Similarly, the first reviewer suggested we add a section of concluding remarks. As such, we have revised Section 4.3 of the discussion which was titled "A global array of deep sea landers", which is now titled Section 5: Concluding remarks. This section now reads:

"5.0 Concluding remarks Ocean deoxygenation is a global concern, with changes in oxygen conditions potentially impairing the productivity of continental shelves and margins that support important ecosystem services and fisheries. Nanolanders provide a powerful tool to examine short-term, fine-scale fluctuations in nearshore dissolved oxygen and other environmental parameters, and associated ecological responses that are rarely recorded otherwise. Oxygen variability was strongly linked to tidal processes, and contrary to expectation, oxygen variability did not decline linearly with depth. Depths of 200 and 400 m showed especially high oxygen variability which may buffer communities at these depths to deoxygenation stress by exposing them to periods of relatively high oxygen conditions across short timescales (daily and weekly). Despite experiencing high oxygen variability, seafloor communities showed limited responses to changing conditions at these short time-scales. However, our deployments did not capture any large acute changes in environmental conditions, that may elicit stronger community responses; future studies using this platform could allow for such observations.

Nanolanders provide a cost-effective and easily deployable tool for studying local conditions throughout the world. Many of the areas where large decreases in oxygen have been observed occur in developing countries, such as along the western and eastern coast of Africa (Schmidtko et al. 2017). Large oxygen losses have also been observed in the Arctic (Schmidtko et al. 2017), where the seafloor habitat is understudied. Due to their compact design, small landers such as DOV BEEBE can provide easy access to nearshore, deep-sea ecosystems and could expand the capacity of developed and developing countries to monitor and study environmental changes along their coastlines. We found that the Nanolander performed well and reliably over the course of the deployments, and allowed us to study seafloor community responses within the context of short-term environmental forcing. For continental margins and seafloor habitats, a global array of Nanolanders, similar in scope to the Argo program, could be envisioned. These would provide coupled physical, biogeochemical, and ecological measurements, which would greatly expand our understanding of temporal and spatial heterogeneity in nearshore deep-sea ecosystems and seafloor community sensitivity to environmental change. ”

As the paper claims to introduce a novel lander-technology, I would have wished to find a brief review of similar already existing systems. The authors mention papers by Jamieson et al. but do not provide details. Please add a few lines highlighting where your system goes beyond existing systems.

RESPONSE: Thank you for highlighting this omission. We propose adding the following text within the manuscript introduction:

"Untethered instrumental seafloor platforms, sometimes called "ocean landers", have a long and rich history (Ewing and Vine 1938, Tengberg et al. 1995). These vehicles are self-buoyant, with an expendable descent anchor that is released by surface command or on-board timer, allowing the vehicle to float back to the surface. The novel design aspects of the Nanolander include the use of plastic spheres for both instrument housing and flotation, which allow the vehicle to be smaller and lighter. Previous generation

landers, such as the landers used for the DEEPSEA CHALLENGE Expedition (Gallo et al. 2015) used syntactic foam for flotation, which is more expensive and requires a metal support frame, thus increasing weight. While glass spheres have a deeper maximum operational depth (to 10 km), the use of glass-filled polyamide spheres in the Nanolander has certain advantages including for machining, threading and bonding. These plastic spheres have a maximum operational depth of 2 km, and decrease the price point compared to using glass spheres. The compact, modular, lightweight design of the Nanolander is also a novel element, as most deep-sea landers require an A-frame and winch to deploy. Additionally, new electronic systems are used in the Nanolander, which have been developed to fit within the spatial constraints of the plastic spheres. The Nanolander is configured to collect paired physical, biogeochemical, and biological data in the deep-sea over multiple days, which is a rarity except for in areas with developed ocean observatories."

Beside oxygen, other parameters were measured (temperature, pH, saturation state of aragonite/calcite) but these were hardly mentioned in the discussion section although e.g. pH in respiration physiology is very important. Please clarify why these parameters were not further included in the interpretation of the data set.

RESPONSE: We agree that temperature, pH, and saturation state of aragonite/calcite are important parameters to consider as well, and for this reason have included this data in the description of environmental variability across deployments. However, since oxygen and carbonate chemistry parameters co-vary (Alin et al. 2012), we are not able to decouple these in our interpretation of community-level differences in the dataset. Mobile, adult fishes are not as susceptible to low pH as to changes in oxygen availability (Melzner et al. 2009, Kroeker 2013), and we therefore make the assumption that respiratory demands play an important role in driving community responses. Additional deployments, including longer-term deployments and repeat deployments at the same sites, would provide the opportunity to perform more in-depth analyses into the roles of different environmental drivers in giving rise to specific community responses. Additionally, we do not think that decreasing pH with depth is responsible for the observed shift from vertebrates to invertebrates, since many of the invertebrates at these deeper depths are calcifers, which should be more sensitive to low pH conditions.

Further comments and edits: Line 24: please explain "phest"

RESPONSE: Changed to "estimated pH" instead of "pHest". This is a calculated pH using empirical relationships derived for this region in Alin et al. (2012). Within the manuscript, pHest is defined as: "pHest is estimated pH, calculated using empirical relationships from Alin et al. (2012)."

Line 67-72 in this context eddy correlation techniques could be mentioned

RESPONSE: We have added: "Eddy correlation techniques are also used to measure non-invasive oxygen fluxes at the seafloor."

Line 108: suggest to use only metric units of m or cm instead of ft

RESPONSE: Only metric units are now included. This section now reads, "DOV BEEBE stands 1.6 m tall and is 0.36 m wide and 0.36 m deep... Within the frame sit three plastic spheres that are 25.4 cm in diameter.... When DOV BEEBE is deployed, the vertical distance from the base of the Nanolander to the seafloor is ∼51 cm (Fig. 1B)."

Figure 1, suggest to include a more detailed technical drawing of the lander (i.e. better version of Fig. 1A) where the different major components are labeled with numbers which can referred to in the main text. Figure 1D is not really providing any additional information and could be omitted. Please provide in the final version of the MS the figures in sufficient resolution.

RESPONSE: We have provided an additional technical drawing of the lander as New Figure 2. This could be included as a supplement or within the manuscript, depending on the preference of the reviewer and editor. The caption that goes with the labels is as follows:

New Figure 2: The primary components of the Nanolander design that yield it's relatively small size and lightweight are: 1) Spectra Lifting bale; 2) HDPE centerplate; 3) 25.4 cm in diameter polyamide spheres stacked top, middle and bottom, 4) sphere retainer (3 ea); 5) auxiliary 17.8 cm in diameter flotation sphere; 6) oil-filled LED light, one port side, one starboard side; 7) Seabird MicroCAT-ODO in the lower payload bay; 8) central fiberglass frame; 9) stabilizing counterweight; 10) anchor slip ring; 11) expendable iron anchor (bar bell weights); 12) burnwire release and mount, one port side, one starboard side; 13) Edgetech hydrophone for acoustic command and tracking; 14) HDPE side panel, one port side, one starboard side; and 15) surface recovery flag. Not shown: drop arm on front.

We have kept Figure 1D within the manuscript figure because it showcases that the Nanolander fits exactly in a horizontal configuration within a Panga, or small boat. These types of boats are readily available all around the world and are typically used by artisanal and recreational fishermen. Thus being able to transport, deploy, and recover the Nanolander in these boats increases the potential for using this scientific platform in many regions of the world, both developed and developing.

Line 111: "glass filled" sounds a bit odd; do you mean glass-spheres housed by polyamide protective shells?

RESPONSE: The plastic spheres are an injection-molded polyamide (nylon) with 30% glass fibers for additional strength. The composite approach is similar to FRP (Fiber Reinforced Plastic), or "Fiberglass", but using nylon as the binder, not polyester epoxy. We have clarified this in the manuscript and modified it to read, "Within the frame sit three plastic spheres that are 25.4 cm in diameter; the spheres are made of injection-molded polyamide with 30% glass fibers for additional strength."

Line 122: "The power supply for the BART board is housed in the upper sphere", together with the Bart board?

RESPONSE: This is correct. A battery pack supplies the power to the BART board,

and is housed in the upper sphere together with the BART board. We have clarified this to read, "The power supply for the BART board is housed in the upper sphere with the BART board."

Line 123: what would be the maximum deployment time of DOV Beebe with the given battery systems?

RESPONSE: In this study, the main technological limitation we ran into was limited battery capacity to power the LED lights; all other elements would have allowed for longer sampling (camera battery and memory, SBE MicroCAT battery and memory). The basic Nanolander itself can stay in situ for periods of 2 years, perhaps longer. However, we were only able to provide sufficient power to the LED lights for a maximum period of 14 days at the selected sampling frequency. It is unclear why during certain deployments, we experienced inconsistent power capacity for the LED lights.

Line 131: would be nice if especially details of the camera system could better show up in the improved version of Figure 1A

RESPONSE: The camera system was designed by Ronan Gray (SubAqua Imaging Systems, San Diego, CA) and William Hagey (Pisces Design, La Jolla, CA) and uses a modified Mobius Action Camera with a time-lapse assembly. They provided a 10 page manual to accompany the camera system, which could be included as an additional supplement to the paper, if the reviewer and editor would like. We have attached the manual for your consideration, titled "The SphereCam Manual."

Line 153: please use metric units

RESPONSE: Changed to: "DOV BEEBE is positively buoyant in water, and is deployed with ~18 kg of sacrificial iron weights."

Line 178: I think there is no need to use the word "high-frequency" (it's rather a matter of the perspective whether 5 min sampling rate is high-frequency or not)

RESPONSE: The word "high-frequency was removed". The sentence now reads,

"Upon recovery of the Nanolander, time-series data from the MicroCAT were analyzed to assess how environmental variability (O2, T, salinity) changes with depth."

Line 194 please describe spiciness in a bit more detail, it's likely not common to everybody

RESPONSE: We have further clarified this in the text, which now reads: "Spiciness, the degree to which water is warm and salty, is a state variable that is conserved along isopycnal surfaces (Flament 2002) and can be used as a tracer for PEW (Nam et al. 2015). We calculated spiciness using the "oce" R package (Kelley and Richards 2017) and examined how oxygen concentration varies with temperature and spiciness across depths and deployments. Spiciness is used to examine differences in spatial variation between watermasses, which otherwise may not be apparent using isopycnal surfaces because the effects of warm temperature and high salinity cancel each other out. "Spicier" water is warmer and saltier."

Line 309 deconstructed time series - please explain in more detail

RESPONSE: This was further clarified in the text, which now reads: "When the time series were decomposed into their additive components (i.e. daily trend, underlying trend, and random noise), time series for all depths showed a clear diurnal and semi-diurnal signal (Supplement 1B)."

We have also added a citation for the R package used for the decomposition in the Methods section, "The oxygen time series for each deployment was also decomposed using the "stats" package (R Core Team 2019) to look at the trend, daily, and random signals that contribute to the overall data patterns."

Figure 4: the labels for "day" and "night" are difficult to read – please enlarge

RESPONSE: We have enlarged the label sizes for day and night. The updated figure with the labels enlarged is attached below (labeled Figure 4 Updated).

Line 448: I am not sure whether the statement "At âĹij200 m, oxygen, temperature,

and pH exhibited high variability (Fig. 2), greater at times than the variability observed at 100 m." is correct for temperature – please check.

RESPONSE: Thank you for pointing this out. I have taken a close look at Table 1 and inspected the overall ranges and CVs for the different deployments. For D200-DM, variability for temperature was higher than during D100-DM-Spr, based on the overall range and CV, but lower than during D100-DM-Fall. Temperature variability for D200-LJ-2 was very similar to temperature variability for D100-DM-Spr but also lower than D100-DM-Fall, based on overall range and CV (Table 1). D200-LJ-1 has lower temperature variability than both D100-DM-Spr and D100-DM-Fall. Since temperature variability was only higher at 200 m than at 100 m in one case, I will clarify the sentence to read: "At ≈200 m, oxygen, temperature, and pH exhibited high variability (Fig. 2), greater at times than the variability observed at 100 m for oxygen and pH."

Although the Figure 2 is quite attractive and informative, especially for the discussion section, when environmental variability is discussed additional Box plots might be helpful to elucidate the differences between the different deployments (i.e. depths).

RESPONSE: Thank you for your positive feedback on Figure 2. We have added additional violin plots to a new figure (New Figure 7), which additionally addresses question 4 raised by Reviewer 1. We provided the following description about this new figure:

To compare our high-frequency measurements to a longer-term dataset, we incorporated data from a nearby CalCOFI station to provide additional context to our results. We relied on data from CalCOFI Station 93.3 28 since it was the closest station to our deployments which sampled the full upper water column down to 500 m... We then used all available CTD casts for Station 93.3 28, which represented data from 65 CalCOFI cruises during the time-period between July 2003 and November 2019, and looked at how the overall variability in environmental conditions across this longer (∼16 year) time-period, compares to the overall variability in environmental conditions across our shorter (∼3-week deployments). These results are presented in a new fig-

ure labeled New Figure 7. This figure shows how the mean, variance (indicated using +/- 1SD and +/- 2SD), and coefficient of variation (CV) for temperature and oxygen change across the upper 500m of the water column at Station 93.3 28 (Panels A-D). This figure also selects data from specific depths that relate to our targeted deployment depths (100, 200, 300, and 400 m), and shows how the variance distribution in temperature and oxygen across our ~3 week deployments compares to the observed variance at these depths over ~16 years of CalCOFI cruise measurements (Panels E-H). Additionally, we have looked for evidence of linear changes in temperature or oxygen at our targeted deployment depths (100, 200, 300, and 400 m) at CalCOFI Station 93.3 28 (Panels I-J) as additional context for longer-term change. We hope that this added analysis helps frame our results regarding variability over short timescales within the context of variability over interannual and multidecadal timescales.

Line 476 Turbidity can be related to local hydrodynamics caused by the energy dissipation of incipient internal tides at sloping boundaries affecting the suspension, transport and deposition of food particles. If you are interested, please see e.g. Mosch et al. (2012) Factors influencing the distribution of epibenthic megafauna across the Peruvian oxygen minimum zone. Deep-Sea Research I 68 (2012) 123–135 and references therein.

RESPONSE: Thank you for pointing us to this interesting study. Similarly, Reviewer 1 also raised the importance of internal tides breaking on the margin as a source of environmental variability that may also explain the high turbidity conditions observed during our 300 m deployment (see RC1 comment 6).

In section 4.1 of our discussion, we have added suggested references from both reviewers, "The high turbidity observed at this depth may be due to shoaling and breaking nonlinear internal waves that can form bottom nepheloid layers (McPhee-Shaw 2006, Boegman and Stastna 2019). On the Peruvian margin, energy dissipation from tidally-driven internal waves have been shown to influence the distribution of epibenthic organisms by increasing suspension, transport, and deposition of food particles (Mosch

et al. 2012). High turbidity conditions have also been observed during two separate ROV dives at ∼340 m off Point Loma (unpublished, NDGallo), suggesting high turbidity conditions may be the norm at these depths on the upper slope in the SCB."

We thank the reviewer for their time and appreciate the improvements to the manuscript that have resulted from making these changes.

Please also note the supplement to this comment:
https://www.biogeosciences-discuss.net/bg-2020-75/bg-2020-75-AC3-supplement.pdf

—————————————————————

**Nanolander**

**Fig. 1.** New Figure 2

[Figure]

**Fig. 2.** Figure 4 Updated

[Figure]

**Fig. 3.** New Figure 7

**Supplement:**

The photos that follow explain how to operate the TIMELAPSE ASSEMBLY.

**Figure 1 Overview. The camera is located in the lower hemisphere and the components for the lights are located in the upper hemisphere.**

The Sphere-Cam uses our existing time-lapse assembly.  We tested many small cameras before we choose the one that we use in it.  The model is the Mobius Action Camera (https://www.mobius-actioncam.com/).  The camera has become very popular with a range of users in particular, Radio Control airplane and drone users.  There are some resources online that will help if you ever need them:

- Camera related software and owner's manual can be used when the camera is connected to a computer via the USB port.  For the latest software check here:  https://www.mobius-actioncam.com/downloads-info/
- To access the cameras settings download the program Msetup.zip
- There is an excellent support forum for the camera here: https://www.rcgroups.com/forums/showpost.php?p=25170910&postcount=4

The USB port and MicroSD card have been positioned within the camera sphere in such a way that you should be able to do everything needed without ever having to remove the camera.

[Figure]

**Camera Hemisphere**

Figure 2  The Timelapse camera sphere data can be downloaded using the USB port or the Memory Card can be removed and replaced.  The battery camera can be charged using the charging port next to the dip-switch.   The dip-switch in the photo is in the 1101 (Every 10 min, 10 sec of  Video).

[Figure]

1. The time lapse intervals can be set by adjusting the dip-switches.  Note that power must be cycled to allow the changes in settings to be read into memory.
    a. The numbers in the table below are read from left to right corresponding with switch #1 through #4.  Switch #5 is NOT USED.
    b. A "0" indicates that the switch should be in the "off" position.  A "1" indicates that the switch should be in the "on" position.  A switch is "on" when it is slid all the way up, next to the number on the body.
    c. **EXAMPLE:**  1000 means switch 1 on, switches 2,3 & 4 are off.

| Option | DIP-SWITCH POSITIONS | Function |
|---|---|---|
| 0 | 0000 | Video |
| 1 | 1000 | 30-second Photo |
| 2 | 0100 | 1-minute Photo |
| 3 | 1100 | 2-minute Photo |
| 4 | 0010 | 5-minute Photo |
| 5 | 1010 | 10-minute Photo |
| 6 | 0110 | 15-minute Photo |
| 7 | 1110 | 30-minute Photo |
| 8 | 0001 | 60-minute Photo |
| 9 | 1001 | Every 5  Mins, 1 Mins Video |
| 10 | 0101 | Every 1/2 Hrs, 5 Mins Video |
| 11 | 1101 | Every 10 min, 10 sec of  Video |
| 12 | 0011 | Every 20 min, 20 sec of Video |
| 13 | 1011 | Every 30 min, 30 sec of Video |
| 14 | 0111 | Every 60 min,  60sec Video |
| 15 | 1111 | DORMANT |

[Figure]

**Trigger LED**

**Dummy Plug. Attach when camera is not in use and not connected to connector hemisphere.**

Figure 3  The camera has a small Trigger LED that activates a photo-relay in the connector hs.  Do not block the LED. The magnetic release attached to the connector hs is connected to the camera via a RCA cable.  Replace the dummy plug when not in use.

[Figure]

1134 Opal St, San Diego, CA 92109, USA – Ph: (858) 414 0383 – Fax (858) 4834884 – www.subaquaimaging.com

Figure 4  The window port located on the bottom of the camra hemisphere.  The red mark on the lens indicates the upright positon.

**Connector Hemisphere**

The Connector Hemisphere has five ports:
1. 12 VDC power for lights only (the camera has it's own battery within the camera hemisphere).
2. Vacuum Port
3. Magnetic Switch (removing the brightly colored cap activates the camera).
4. LED Light cable
5. LED Light cable

[Figure]

The LED drivers are the largest components in the connector sphere.  There is one for each light and they are powered by the external 12 VDC power source connected to the hemisphere by the power connector.  The positive leads are routed through a photo-relay which closes and allows power from the power connector to flow to them when the LED in the camera hemisphere is activated.

A 5-pin terminal strip is used to connect the output from the LED drivers to each of the LED connectors.

The positive lead of the 12VDC power runs through the 5-position terminal strip and supplies the relay and the relay switch.

A 2-postion strip located next to the photo-relay board connects the negative lead from the 12VDC connector to the LED drivers and the photo relay board.

Note the RED LED above and to the right of the *"12 VDC input for relay"*.  This LED should be ON when the external 12 VDC power is properly attached to the hemisphere, regardless of whether the magnetic switch is activated or not.

[Figure]

**Figure 5  Note the RED LED above and to the right of the *"12 VDC input for relay"*.  This LED should be ON when the external 12 VDC power is properly attached to the hemisphere, regardless of whether the magnetic switch is activated or not.**

---

## Author Response (AR1)

Dear Dr. Tina Treude and Dr. Marilaure Grégoire,

First, we respond specifically to the points raised by Dr. Treude, and then point-by-point to the reviews, and then list the major changes. Our responses are indicated in bold. A version of the manuscript with changes tracked is included at the end of this document. We apologize for the slight delay in the resubmission of our revised manuscript and thank you for your oversight of our manuscript. We believe that the thoughtful feedback from the reviewers has resulted in an improved manuscript and we hope that these changes make the manuscript acceptable for publication in *Biogeosciences*.

Sincerely,

Dr. Natalya Gallo, on behalf of co-authors
* * *
Comments from Dr. Treude:

"I support the suggestion to include Fig 2 (the detailed technical drawing) in the main manuscript."

**We have included the technical drawing as Figure 2.**

"I agree to keep Fig. 1D to show how the lander can be deployed from small boats."

**Thank you, we have kept this image in.**

"I support the suggestion from the external comment to include a supplementary table with fish species observed and min/max depths."

**We have added this table as Supplement 1C.**

"Please make sure to elaborate on the battery run time depending on the measurement settings as asked by reviewer #2. I believe this is an information many readers working in this field will appreciate."

**We have elaborated on this in two sections. First, in the results:**

**"Memory and power capacity often limit deployment times for long-term, deep-sea deployments. In this study, the main technological limitation we ran into was limited battery capacity to power the LED lights. As opposed to 8 hours of estimated LED performance time, field performance ranged from 2.2 to 6.6 hours total time, which meant that the total time of biological data collection was shortened and ranged from 5.5 to 16.5 days, respectively (Table 1). Memory and power were not issues for the camera system; the 128 GB micro SD card was cleared and the battery pack was fully recharged following each deployment. Video quality was high enough to allow species-level identifications and the light from the LEDs was sufficient to light the field of view (Fig. 1F and G). The SeaBird MicroCAT-ODO also performed without any issues and had sufficient battery and memory capacity for all deployments. If not for power limitations to the LED lights, the camera system and SBE MicroCAT would have allowed for longer sampling (~1 month and potentially longer). The basic Nanolander itself can stay *in situ* for up to two years. Detailed descriptions of Nanolander performance can be found in Gallo (2018)."**

**Additionally, in the new concluding remarks section, we added the following:**

**"Our deployment lengths were limited by battery capacity to power the LED lights; all other elements would have allowed for longer sampling duration. Specific ways to extend future deployment lengths are currently being explored and include: using higher efficacy LEDs, integrating additional batteries to power the LED lights into newly devised Nanolander side pods, improving circuit performance that powers the LED lights by using new camera controllers and solid-state relays, and using low-light cameras, such as the Sony 7S II, which reduce the light required to illuminate the field of view. Longer deployment lengths would be advantageous for capturing ecosystem responses to environmental variability across time-scales (hours to months)."**
* * *
Responses to Reviewers:

Reviewer 1 (SungHyun Nam)

1) What is the bottom topography around the Nanolanders? Table 1 well summarizes the seven deployments including information on deployment location and depth. But, 'Scripps Coastal Reserve', 'Del Mar Steeples Reef' with latitudes/longitudes and bottom depth are not enough information for readers (particularly someone who is not familiar to the region) to figure out local bathymetric features, where outer shelf and upper slope are located/ranged/shaped, seafloor area exposed to different oxygen conditions, and so on. It is important to give details of the bathymetry around the de- ployment sites highlighting the key information as mentioned above. This would also be helpful for better discussing physical drivers of the oxygen variability. Thus, I would like to suggest to add one figure (or incorporated into Figure 1) showing compact and easy to understand map of the local bathymetry along with the deployment locations."

**We have created a new figure (Fig. 3) which clearly shows the deployment locations (green diamonds) in relationship to local bathymetric features. Despite relatively close spatial proximity between the Scripps Reserve and Del Mar deployment sites (~10 km), there are important bathymetric differences. The Scripps Coastal Reserve deployment sites are positioned close to a submarine canyon feature (the La Jolla canyon), while the Del Mar Steeples Reef deployment sites are on a gradually sloping margin. Additionally, we have added the locations of nearby CalCOFI stations (93.3 26.7 and 93.3 30) (black circles) and have included data from CalCOFI station 93.3 30 to provide additional context regarding variability over longer timescales.**

**The manuscript now reads:**

**"Seven deployments were conducted during the study period, ranging from 15-35 days, and at targeted depths of 100-400 m (Table 1). Two early deployments (D200-LJ-1 and D200-LJ-2) were done near the Scripps Reserve off La Jolla, CA and five subsequent deployments (D100-DM-Fall, D200-DM, D300-DM, D400-DM, D100-DM-Spr) targeted a nearby rockfish habitat – the Del Mar Steeples Reef, CA (Fig. 3). Despite relatively close spatial proximity (~10 km), the local bathymetry differed between the LJ and DM deployments (Fig. 3); the LJ deployments were close to a submarine canyon feature, while the DM deployments were on a gradually sloping margin."**

2) There is no summary/concluding remarks/conclusion in the manuscript. Substantial conclusions are reached but they are not presented as a separate section. Thus, I would like to suggest to add Section 5 to conclude or summarize the materials.

**We have modified what was previously section 4.3 of the discussion, titled "A global array of deep sea landers", added additional content summarizing the findings of the study, and titled this section: "Section 5: Concluding remarks". This section now reads:**

**"Ocean deoxygenation is a global concern, with changes in oxygen conditions potentially impairing the productivity of continental shelves and margins that support important ecosystem services and fisheries. Nanolanders provide a powerful tool to examine short-term, fine-scale fluctuations in nearshore dissolved oxygen and other environmental parameters, and associated ecological responses that are rarely recorded otherwise. Oxygen variability was strongly linked to tidal processes, and contrary to expectation, high-frequency oxygen variability did not decline linearly with depth. Depths of 200 and 400 m showed especially high oxygen variability and seafloor communities at these depths may be more resilient to deoxygenation stress because animals are exposed to periods of reprieve during higher O2 conditions and may undergo physiological acclimation during periods of low O2 conditions at daily and weekly timescales. Despite experiencing high oxygen variability, seafloor communities showed limited responses to changing conditions at these short timescales. However, our deployments did not capture any large acute changes in environmental conditions that may elicit stronger community responses; future studies using this platform could allow for such observations.**
**The Nanolander DOV BEEBE is configured to collect paired physical, biogeochemical, and biological data in the deep-sea over multiple days, which is a rarity except for in areas with developed ocean observatories. We found that DOV BEEBE performed well over the course of the deployments, and allowed us to study seafloor community responses to short-term environmental forcing. Our deployment lengths were limited by battery capacity to power the LED lights; all other elements would have allowed for longer sampling duration. Specific ways to extend future deployment lengths are currently being explored and include: using higher efficacy LEDs, integrating additional batteries to power the LED lights into newly devised Nanolander side pods, improving circuit performance that powers the LED lights by using new camera controllers and solid-state relays, and using low-light cameras, such as the Sony 7S II, which reduce the light required to illuminate the field of view. Longer deployment lengths would be advantageous for capturing ecosystem responses to environmental variability across time-scales (hours to months).**
**Many of the areas where large decreases in oxygen have been observed occur in developing countries, such as along the western and eastern coast of Africa (Schmidtko et al. 2017). Large oxygen losses have also been observed in the Arctic (Schmidtko et al. 2017), where the seafloor habitat is understudied. Due to their compact design, small landers such as DOV BEEBE can provide a cost-effective and easily deployable tool for studying nearshore, deep-sea ecosystems and thus expand the capacity of developed and developing countries to monitor and study environmental changes along their coastlines. For continental margins and seafloor habitats, a global array of Nanolanders, similar in scope to the Argo program, could be envisioned. These would provide coupled physical, biogeochemical, and ecological measurements, which would greatly expand our understanding of temporal and spatial heterogeneity in nearshore deep-sea ecosystems and seafloor community sensitivity to environmental change."**

3) To give proper credit to related work, I would like to suggest to use '13CW', name of specific water mass linked to the deoxygenated water, instead of its locally defined water types, Pacific Equatorial Water (PEW) although previous works used the terms PEW. Based on recent work (Zachary et al., 2020; "The role of water masses in shaping the distribution of redox active compounds in the Eastern Tropical North Pacific oxygen deficient zone and influencing low oxygen concentrations in the eastern Pacific Ocean" published in Limnology and Oceanography as of 06 February 2020), two water masses – 13CW and deeper North Equatorial Pacific Intermediate Water (NEPIW) act as the two Pacific Equatorial source waters to the California Current System corresponding to upper and lower PEW at isopycnals of 26.2-26.8 kg m-3 when defined locally. Here, the relevant water mass seems to be 13CW (upper PEW), and not NEPIW (lower PEW).

**Thank you for drawing our attention to this new reference. We have added the following clarification in the manuscript:**

**In Section 2.2: "Previous studies have found that changes in oxygen and pH in the Southern California Bight are associated with changes in the volume of Pacific Equatorial Water (PEW) transported in the California Undercurrent (Bograd et al. 2015, Nam et al. 2015). PEW is characterized by low oxygen, warm, high salinity conditions, and is composed of two water masses, the 13°C water mass (13CW) and the deeper Northern Equatorial Pacific Intermediate Water Mass (NEPIW) (Evans et al. 2020)."**

**To maintain consistency with the nomenclature in the reference, we have kept the use of "PEW" in cases where it directly refers to the results of a study. For example, "Previous studies have found that changes in oxygen and pH in the Southern California Bight are associated with changes in the volume of Pacific Equatorial Water (PEW) transported in the California Undercurrent (Bograd et al. 2015, Nam et al. 2015)."**

**But have otherwise used the 13CW terminology when referencing this water mass in the paper.**

4) The observed oxygen variability over short time scales was compared with multi- decade-long deoxygenation or long-term trends/shifts reported in Bograd et al. (2008) and McClatchie et al. (2010). However, it was not discussed in comparison to inter-annual oxygen variability in the region. Does the period of data collection from August 2017 to March 2018 correspond to normal or more likely abnormal (El Niño/La Niña) year? My suggestion is to provide discussions on the observational results in terms of significant local interannual oxygen variability in association with such large-scale condition presented in Nam et al. (2011; "Amplification of hypoxia and acidic events by La Niña conditions on the continental shelf off California" published in Geophysical Research Letters as of 23 November 2011).

**Thank you for this suggestion. Indeed, we were interested in comparing how short-term variability compares to longer-term variability driven both by interannual and multidecadal changes as one of the objectives of this research. To compare our high-frequency measurements to a longer-term dataset, we incorporated data from a nearby CalCOFI station to provide additional context to our results.**

**We have added the following to the manuscript:**

**"2.3 Short-term oxygen variability in the context of longer trends**

To examine $O_2$ variability over shorter (i.e. daily and weekly) timescales compared to longer (i.e. seasonal, interannual, multidecadal) timescales, we compared our Nanolander results with the annual rates of oxygen loss reported for the SCB nearshore region (Bograd et al. 2008) as well as CTD casts from nearby CalCOFI station 93.3 28 (Fig. 3). CalCOFI station 93.3 26.7 was also nearby, but was too shallow for comparison with the Nanolander deployments (Fig. 3). Quality controlled CTD casts from station 93.3 28 were available for a ~16-year period (Oct 2003-November 2019), representing data from 61 cruises (calcofi.org). CTD data were used to examine characteristics of variability of temperature and oxygen through the water column, including mean conditions, the standard deviation, and the coefficient of variation across the 16-year period of quarterly samples. Oxygen data at 100, 200, 300, and 400 m were extracted to compare the distribution of observations across this 16-year period with the high-frequency measurements from the ~3-week Nanolander deployments. Additionally, we tested for significant linear trends in temperature or oxygen at 100, 200, 300, and 400 m, to examine recent (2003-2019) warming and deoxygenation trends at the CalCOFI station closest to the Nanolander deployments."

We have then provided these additional CalCOFI data as context for the Nanolander data within the revised discussion section: 4.1 Comparing oxygen variability: Short-term to long-term trends. The new additions read:

"CalCOFI data from nearby station 93.3 28 provides additional context on the characteristics of oxygen variability across seasonal and interannual timescales. When temperature and oxygen profiles from ~16 years of quarterly CalCOFI cruises are examined, we see that the highest temperature variability occurs in the upper water column (<50 m) and variability below ~150 m is relatively low (Fig. 7A,B). In contrast, absolute oxygen variability (i.e. standard deviation) is greatest between 50-150 m (Fig. 7C), and the coefficient of variation for oxygen (CV) actually increases below 100 m (Fig. 7D).

Comparing our high-frequency Nanolander deployment results to oxygen measurements across these ~16 years of quarterly CalCOFI cruises, we observe that the range in oxygen measurements at ~100 m, 300 m, and 400 m only captured a small portion of the variability measured across the ~16 year time period. In contrast, for the ~200 m deployments, a significant fraction of the variance over seasonal and interannual time-periods was captured by the short-term deployments (Fig. 7E-H). Oxygen variability in the SCB is also affected by the El Niño Southern Oscillation (ENSO), with oxygen conditions lower during La Niña periods (Nam et al. 2011). During the Nanolander deployments (August 2017-March 2018), the monthly Niño-3.4 index was always negative (-0.21 to -1.04; cpc.ncep.noaa.gov) but weaker than the La Niña conditions described in Nam et al. (2011). Our deployments, therefore captured a neutral ENSO/weak La Niña state. Interannual variability due to ENSO is captured in the data distribution from the CalCOFI cruises.

In recent years (2003-2019), at the CalCOFI station closest to the Nanolander deployments (93.3 28), no significant linear deoxygenation trends were detected at 100 or 200 m, but significant deoxygenation trends were detected for 300 and 400 m (300 m: LR, $R^2 = 0.10$, p < 0.001; 400 m: LR, $R^2 = 0.21$, p < 0.001) (Fig. 7J). No significant warming trends were detected at these depths during this period (Fig. 7I). At 300 m, oxygen declined by 0.89 µmol kg$^{-1}$ year$^{-1}$ during the ~16 year time period, leading to a total oxygen loss of 14.25 µmol kg$^{-1}$ across the timeseries, and at 400 m oxygen declined by 0.94 µmol kg$^{-1}$ year$^{-1}$, leading to a total oxygen loss of 15.11 µmol kg$^{-1}$ over the ~16 years. Comparatively, the range of oxygen conditions experienced

over the ~3-week Nanolander deployment was ~19 µmol kg$^{-1}$ at 300 m and ~17 µmol kg$^{-1}$ at 400 m.”

A new figure was also produced and the figure caption reads:

“**Figure 7: Comparing short-term environmental variability from *DOV BEEBE* deployments to longer-term trends using CTD casts at nearby CalCOFI station (93.3 28.0). Mean temperature (A) and oxygen (C) conditions through the water column (0-500 m) using CalCOFI CTD casts from Oct 2003-November 2019; light and dark colors indicate the variance around the mean and represent +/- 1 and 2 SD, respectively. Panels B and D show how the coefficient of variation (CV) for temperature and oxygen changes through the water column. Dotted lines in A-D indicate 100, 200, 300, and 400 m depths, and data are extracted for these depths for E-J. In E-H, violin plots show data distribution of oxygen measurements from ~16 years of CalCOFI quarterly cruises compared to ~3 week Nanolander deployments at 100 m (E), 200 m (F), 300 m (G), and 400 m (H). Violin plots show the mean +/- 1 SD (white) and +/- 2 SD (black). Panels I and J examine changes in temperature (I) and oxygen (J) conditions through time at 100, 200, 300, and 400 m. Dotted lines indicate non-significant linear relationships; solid lines indicate significant trends (p < 0.05).”**

5) What are depths of thermocline/oxycline (any strong vertical temperature/oxygen gradient close to 200 m?) and their sectional structures across the shelf-slope? It would be helpful to check the cross-sectional structures of water temperature and dis- solved oxygen across the shelf and slope at a given time, e.g., see Figure 2 of Nam et al. (2011) but focusing on the deeper area (over the slope). Both mean and standard deviation to the mean, thus the CV of the temperature/oxygen can be partly explained from its vertical (and horizontal) gradient. My question is whether relatively high CV is due to strong vertical (or horizontal) gradient of temperature and oxygen (thermocline and oxycline depths). Also, how the structures are different from spring (D100-DM-Spr) vs. fall (DM100-DM-Fall)? It would also be relevant to high turbidity condition around 300 m as the internal waves/internal tides break and enhance the mixing (to resuspend the sediment) when and where the isopycnals (isotherms) touch the bottom (see the comments #6 below for details).

**To look at patterns in cross-sectional structures of water temperature and dissolved oxygen across the shelf and slope, and to look at how these spatial patterns change seasonally, we extracted data from CalCOFI stations 93.3 28, 93.3 30, 93.3 35, and 93.3 40 and examined the CTD profiles for these stations during the deployment period. Four cruises were relevant to examine, however, cruise 1802SH was shortened due to the government shutdown and therefore only one of the four stations (93.3 30) was sampled. As such, we focused this additional analysis on just the three cruises (1708SR – August 2017, 1711SR – November 2017, and 1804SH – April 2018).**

**We created a new supplementary figure (Supplementary 1G) in which we show the temperature and oxygen profiles for these four stations across the three relevant cruises. From these profiles, we see that in the spring (April 2018), there is no onshore- offshore gradient, whereas in summer (August 2017) and to a lesser degree in late fall (November 2017), spatial differences in onshore (93.3 28 and 93.3 30) and offshore (93.3 35 and 93.3 40) environmental profiles are evident. These spatial differences are most pronounced in late summer (August 2017). Additionally, in August 2017 there is evidence of some unusual vertical structure in the oxygen profile around ~200 m; both at station 93.3 28 and 93.3 30. Our first deployment (D200-LJ-1) was conducted in late August, so may have captured part of this feature. However, this cannot fully explain the higher**

**variability we observed at 200 m, because our later deployment (D200-DM) was done in mid November, when there is no evidence of unusual vertical structure in the oxygen profile at 200 m for 93.3 28 or 93.3 30. These supplementary profiles, as well as the profiles in Figure 7 do show that the thermocline is steeper and shallower, overall, than the oxycline. We hope these additional datapoints help shed light on the sources of observed variability in our short-term deployments.**

6) As described in Abstract, the high-frequency oxygen variability was strongly linked to tidal processes. But, I do not understand why it is contrary to expectation. As described in Section 1 (Lines 54-57), Section 3 (Lines 308-313), Section 4 (Lines 449-450 and 479-480), and Supplements, diurnal and semidiurnal oxygen variability is noticeable. This is not something unexpected but consistent with previous works reporting oxygen variability in a shallower zone, e.g., Frieder et al. (2012). Importance of tidal processes may also be confirmed from spring-neap cycles or modulations of semidiurnal/diurnal oxygen fluctuations. I could see such a spring (neap) amplification (reduction), for example, from time series plot of D10 - 98 m or D100-DM-Spr in Supplement 1B. Amplitudes of semidiurnal oxygen fluctuations reach up to larger than 20 μmol kg-1 for Days 0-3 and 10-13 (presumably corresponding to spring tide) while smaller than 10 μmol kg-1 for Days 5-8 and 17-20 (presumably corresponding to neap tide). What are CVs for periods of spring vs neap tides? I believed and continue to believe that such high-frequency oxygen variability is relevant to internal tides generated and shoaled at a specific phase of the surface tide in a sloping bottom (even up to the zone as shallow as 15 m) as reported in the region by Nam and Send (2011) and others. It is generally known that the isotherms (so iso-oxygen surfaces) move up and down at high-frequency due to propagation and evolution internal tides and associated shorter period nonlinear internal waves (also termed internal solitary waves). When they shoal and break, turbulent mixing is markedly enhanced often forming bottom nepheloid layer that may account for suspended sediments and the high turbidity condition around 300 m. The bottom nepheloid layer has been presented since McPhee-Shaw (2006; "Boundary-interior exchange: Reviewing the idea that internal-wave mixing enhances lateral dispersal near continental margins" published in Deep Sea Research II: Topical studies in Oceanography as of 20 February 2006), e.g., Boegman and Stastna (2019; "Sediment resuspension and transport by internal solitary waves" published in Annual Reviews of Fluid Mechanics as of 15 August 2018).

**Thank you for raising these points. One of the objectives of this study was to place rates of anthropogenic change within the context of short-term variability that nearshore deep-sea communities are exposed to. These results show that tidally-driven variability is an important source of high-frequency variability to consider, that could either exacerbate or buffer deep-sea communities from changes in mean conditions with climate change. Contrary to the idea that the deep-sea is a stable environment, these results show a substantial amount of environmental variability occurring at short timescales on the upper margin.**

**As suggested by the reviewer, we examined the CVs for the two time periods specified by the reviewer for D100-DM-Spr. The CVs for the two time periods corresponding to the amplified oxygen pattern (Days 0-3 and Days 10-13) were 5.02% and 4.76%, respectively, while the CVs for the two time periods corresponding to the reduced oxygen variability (Days 5-8 and 17-20) were 2.66% and 2.69%, respectively.**

**However, contrary to expectation, a closer examination of the depth time series revealed that there was no consistent evidence that increased tidal amplitude gave rise to increased oxygen variability. We have added additional figures showing the oxygen and depth time series side by side for each deployment in Supplement 1F. For D100-DM-Spr, the initial period of high oxygen**

variability (Days 0-3) corresponded to lower tidal amplitudes, and at the end of the deployment, the period of reduced oxygen variability (Days 17-20) corresponded to high tidal amplitudes.

Furthermore, we have examined this point by calculating the total daily range in oxygen concentration (i.e. daily maximum-daily minimum measurement) and daily range in depths for each deployment and examined if there is evidence of a positive correlation, which would suggest that increases in tidal amplitude (for example during a spring tide) would give rise to increased oxygen variability (as suggested by the reviewer). In fact, we have found the opposite to be true for certain deployments. For deployments D100-DM-Spr, D200-LJ-1, and D200-LJ-2 a negative relationship between tidal variability and oxygen variability can be seen. For D200-DM a weak negative relationship can be seen. For D100-DM-Fall, D300-DM, and D400-DM there is no relationship between tidal variability and oxygen variability. A figure with these results is included in Supplement 1F. These results are quite counterintuitive, and given that our expertise is not in physical oceanography, we do not feel comfortable speculating about the mechanism underlying these observed patterns. It is notable that the deployments that had negative relationships between tidal amplitudes and oxygen variability (Supplement 1F) are the same four deployments that show a clear negative relationship between spiciness and oxygen concentration (Fig. 5). As such, it may have something to do with the California Undercurrent and Pacific Equatorial Water. Perhaps during periods of higher tidal amplitude, the stratification between water masses is slightly reduced, leading to less overall oxygen variability during the tidal cycle. But, again, we do not feel comfortable speculating in the manuscript because this is not our area of expertise. The results are consistent when looking at the relationship between daily patterns of oxygen and tidal variability by calculating the CV, as opposed to looking at the range, so we are confident that the results are robust, though we are not sure how to explain them.

We have added the following section to the manuscript results: "Oxygen variability does not appear to increase with tidal amplitude; for D100-DM-Spr, D200-LJ-1, and D200-LJ-2 the daily oxygen range appears to be negatively correlated with the daily tidal range (Supplement 1F)."

We have also added the suggested references in the discussion in section 4.2:

"The high turbidity observed at 300 m may be due to shoaling and breaking nonlinear internal waves that can form bottom nepheloid layers (McPhee-Shaw 2006, Boegman and Stastna 2019). On the Peruvian margin, energy dissipation from tidally-driven internal waves have been shown to influence the distribution of epibenthic organisms by increasing suspension, transport, and deposition of food particles (Mosch et al. 2012). High turbidity conditions have also been observed during two separate ROV dives at ~340 m off Point Loma (unpublished, NDGallo), suggesting high turbidity conditions may be the norm at these depths on the upper slope in the SCB."

7) Not being a biologist, I do not know in detail how the seafloor communities respond to short-period (mostly diurnal) changes in environmental conditions, but it is convincing that longer time series data are vital for addressing the science issue. My question is why camera sample should be less frequent for longer-term deployment. Is it limited by battery or memory? There would be several technical ways to overcome battery or memory limit. Why not trying new technologies that allow longer-term deployment keeping the same camera (as well as other sensors) sampling frequency.

We have clarified this in the new conclusion section. The new text reads: "Our deployment lengths were limited by battery capacity to power the LED lights; all other elements would have

**allowed for longer sampling duration. Specific ways to extend future deployment lengths are currently being explored and include: using higher efficacy LEDs, integrating additional batteries to power the LED lights into newly devised Nanolander side pods, improving circuit performance that powers the LED lights by using new camera controllers and solid-state relays, and using low-light cameras, such as the Sony 7S II, which reduce the light required to illuminate the field of view. Longer deployment lengths would be advantageous for capturing ecosystem responses to environmental variability across time-scales (hours to months)."**
* * *
Reviewer 2 (Anonymous)

I very much like the idea of the small-sized and hand-operated lander and I fully agree with the need for such systems for the performance of more in situ long-term observations. Yet, given the actual size and weight of the lander, the expression 'Nanolander' seems a bit exaggerated. This is just a personal opinion and is by no means meant to urge the authors to change the name of their system. In this context, the last section of the MS "A global array of deep-sea landers" goes in the same direction and appears a bit superficial with an emphasis on "selling" the system. The authors might consider to rewrite this last section increasing its profoundness.

**We agree in principle that the small benthic lander is not actually "nano", however, Nanolander is now a registered proper name that specifies this scientific platform, so we maintain the name in the manuscript. The term arose to distinguish it from much larger traditional benthic landers.**

**We appreciate the reviewer's perspective that the final section could be further expanded to increase its profoundness. Similarly, the first reviewer suggested we add a section of concluding remarks. As such, we have revised Section 4.3 of the discussion which was titled "A global array of deep sea landers", which is now titled Section 5: Concluding remarks. This section now reads:**

**"Ocean deoxygenation is a global concern, with changes in oxygen conditions potentially impairing the productivity of continental shelves and margins that support important ecosystem services and fisheries. Nanolanders provide a powerful tool to examine short-term, fine-scale fluctuations in nearshore dissolved oxygen and other environmental parameters, and associated ecological responses that are rarely recorded otherwise. Oxygen variability was strongly linked to tidal processes, and contrary to expectation, high-frequency oxygen variability did not decline linearly with depth. Depths of 200 and 400 m showed especially high oxygen variability and seafloor communities at these depths may be more resilient to deoxygenation stress because animals are exposed to periods of reprieve during higher O2 conditions and may undergo physiological acclimation during periods of low O2 conditions at daily and weekly timescales. Despite experiencing high oxygen variability, seafloor communities showed limited responses to changing conditions at these short timescales. However, our deployments did not capture any large acute changes in environmental conditions that may elicit stronger community responses; future studies using this platform could allow for such observations.**
**The Nanolander DOV BEEBE is configured to collect paired physical, biogeochemical, and biological data in the deep-sea over multiple days, which is a rarity except for in areas with developed ocean observatories. We found that DOV BEEBE performed well over the course of the deployments, and allowed us to study seafloor community responses to short-term environmental forcing. Our deployment lengths were limited by battery capacity to power the**

**LED lights; all other elements would have allowed for longer sampling duration. Specific ways to extend future deployment lengths are currently being explored and include: using higher efficacy LEDs, integrating additional batteries to power the LED lights into newly devised Nanolander side pods, improving circuit performance that powers the LED lights by using new camera controllers and solid-state relays, and using low-light cameras, such as the Sony 7S II, which reduce the light required to illuminate the field of view. Longer deployment lengths would be advantageous for capturing ecosystem responses to environmental variability across time-scales (hours to months).**

**Many of the areas where large decreases in oxygen have been observed occur in developing countries, such as along the western and eastern coast of Africa (Schmidtko et al. 2017). Large oxygen losses have also been observed in the Arctic (Schmidtko et al. 2017), where the seafloor habitat is understudied. Due to their compact design, small landers such as DOV BEEBE can provide a cost-effective and easily deployable tool for studying nearshore, deep-sea ecosystems and thus expand the capacity of developed and developing countries to monitor and study environmental changes along their coastlines. For continental margins and seafloor habitats, a global array of Nanolanders, similar in scope to the Argo program, could be envisioned. These would provide coupled physical, biogeochemical, and ecological measurements, which would greatly expand our understanding of temporal and spatial heterogeneity in nearshore deep-sea ecosystems and seafloor community sensitivity to environmental change.”**

As the paper claims to introduce a novel lander-technology, I would have wished to find a brief review of similar already existing systems. The authors mention papers by Jamieson et al. but do not provide details. Please add a few lines highlighting where your system goes beyond existing systems.

**Thank you for highlighting this omission. We have added the following to the introduction:**

**“Untethered instrumented seafloor platforms, sometimes called “ocean landers”, have a long and rich history (Ewing and Vine 1938, Tengberg et al. 1995). These vehicles are self-buoyant, with an expendable descent anchor that is released by surface command or on-board timer, allowing the vehicle to float back to the surface. Autonomous landers have several advantages for deep-sea research, such as lower cost combined with spatial flexibility. Unlike moorings or cabled observatories, small landers (< 2 m high) can easily be recovered using small boats and redeployed to new depths and locations (Priede and Bagley 2000, Jamieson 2016).”**

**We have also tried to highlight the specific novelty of the Nanolander design within Section 2.1 Nanolander development and deployment. This reads:**

**“Autonomous landers have been used successfully to observe abyssal and deep-sea trench communities (e.g., Jamieson et al. 2011, Gallo et al. 2015), however, these landers were large and required a ship with an A-frame and winch to deploy and recover. For this study, the goal was to develop a deep-water lander that could easily be hand-deployed out of a small boat and that was capable of continuously collecting hydrographic and fish and invertebrate assemblage data from near the seafloor for several weeks at a time…. The Nanolander frame is made of marine-grade high-density polyethylene (HDPE) (brand name “Starboard”) and reinforced with fiberglass pultruded channel and angle beams for structure, reducing in-water weight. HDPE has a specific gravity of <1, close to neutrally buoyant. The specific gravity of fiberglass is 2/3 that of aluminium, requiring less flotation to achieve neutral buoyancy. Both plastic materials are impervious to saltwater corrosion…. The novel design aspects of the Nanolander include the use of plastic spheres for both instrument housing and flotation, which allow the vehicle to be smaller**

**and lighter. Previous generation landers, such as the landers used for the *DEEPSEA CHALLENGE* Expedition (Gallo et al. 2015), used syntactic foam for flotation, which is more expensive and requires a metal support frame. While glass spheres have a deeper maximum operational depth (to 11 km), the use of glass-filled polyamide spheres in the Nanolander has machining advantages and decreases the price point. The plastic spheres used for *DOV BEEBE* are pressure-tolerant to 1 km; new spheres with 20% glass content are pressure tolerant to 2 km… This battery system is novel and was developed specifically to fit within the spatial constraints of the sphere and provide high power capacity."**

Beside oxygen, other parameters were measured (temperature, pH, saturation state of aragonite/calcite) but these were hardly mentioned in the discussion section although e.g. pH in respiration physiology is very important. Please clarify why these parameters were not further included in the interpretation of the data set.

**We agree that temperature, pH, and saturation state of aragonite/calcite are important parameters to consider as well, and for this reason have included this data in the description of environmental variability across deployments. However, since oxygen and carbonate chemistry parameters co-vary (Alin et al. 2012), we are not able to decouple these in our interpretation of community-level differences in the dataset. Mobile, adult fishes are not as susceptible to low pH as to changes in oxygen availability (Melzner et al. 2009, Kroeker 2013), and we therefore make the assumption that respiratory demands play an important role in driving community responses.**

**Additional deployments, including longer-term deployments and repeat deployments at the same sites, would provide the opportunity to perform more in-depth analyses into the roles of different environmental drivers in giving rise to specific community responses. Additionally, we do not think that decreasing pH with depth is responsible for the observed shift from vertebrates to invertebrates, since many of the invertebrates at these deeper depths are calcifers, which should be more sensitive to low pH conditions.**

Further comments and edits: Line 24: please explain "phest"

**Changed to "estimated pH" instead of "pHest". This is a calculated pH using empirical relationships derived for this region in Alin et al. (2012). Within the manuscript, pHest is defined as: "pHest is estimated pH, calculated using empirical relationships from Alin et al. (2012)."**

Line 67-72 in this context eddy correlation techniques could be mentioned

**We have added: "Eddy correlation techniques are also used to measure non-invasive oxygen fluxes at the seafloor, however require ROVs or scuba divers to deploy (Berg et al. 2009)."**

Line 108: suggest to use only metric units of m or cm instead of ft

**Only metric units are now included. This section now reads, "DOV BEEBE stands 1.6 m tall and is 0.36 m wide and 0.36 m deep. . . Within the frame sit three plastic spheres that are 25.4 cm in diameter. . .. When DOV BEEBE is deployed, the vertical distance from the base of the Nanolander to the seafloor is ~51 cm (Fig. 1B)."**

Figure 1, suggest to include a more detailed technical drawing of the lander (i.e. better version of Fig. 1A) where the different major components are labeled with numbers which can referred to in the main text. Figure 1D is not really providing any additional information and could be omitted. Please provide in the final version of the MS the figures in sufficient resolution.

**We have provided an additional technical drawing of the lander as Figure 2. The caption that goes with the labels is as follows:**

**"Figure 2: A detailed schematic of the Nanolander *DOV BEEBE* components: 1) Spectra lifting bale; 2) High-density polyethylene (HDPE) centerplate; 3) ~25 cm polyamide spheres stacked top, middle and bottom, see description, Section 2.1; 4) sphere retainer; 5) auxiliary ~18 cm flotation sphere; 6) oil-filled LED lights; 7) Seabird MicroCAT-ODO in the lower payload bay; 8) central fiberglass frame; 9) stabilizing counterweight; 10) anchor slip ring; 11) expendable iron anchor (bar bell weights); 12) burnwire release and mount, one port side, one starboard side; 13) Edgetech hydrophone for acoustic command and tracking; 14) HDPE side panels; and 15) surface recovery flag. Not shown: drop arm on front (see Fig. 1B and 1E)."**

**We have kept Figure 1D within the manuscript figure because it showcases that the Nanolander fits exactly in a horizontal configuration within a Panga, or small boat. These types of boats are readily available all around the world and are typically used by artisanal and recreational fishermen. Thus being able to transport, deploy, and recover the Nanolander in these boats increases the potential for using this scientific platform in many regions of the world, both developed and developing.**

Line 111: "glass filled" sounds a bit odd; do you mean glass-spheres housed by polyamide protective shells?

**The plastic spheres are an injection-molded polyamide (nylon) with 15% glass fibers for additional strength. The composite approach is similar to FRP (Fiber Reinforced Plastic), or "Fiberglass", but using nylon as the binder, not polyester epoxy. We have clarified this in the manuscript and modified it to read, "Within the frame sit three plastic spheres that are 25.4 cm in diameter; the spheres are made of injection-molded polyamide with 15% glass fibers for additional strength."**

Line 122: "The power supply for the BART board is housed in the upper sphere", together with the Bart board?

**This is correct. A battery pack supplies the power to the BART board, and is housed in the upper sphere together with the BART board. We have clarified this to read, "The battery for the BART board is housed in the upper sphere with the BART board."**

Line 123: what would be the maximum deployment time of DOV Beebe with the given battery systems?

**We have added the following information to the manuscript. In the results section 3.1 Nanolander performance:**

**"Memory and power capacity often limit deployment times for long-term, deep-sea deployments. In this study, the main technological limitation we ran into was limited battery capacity to power**

**the LED lights. As opposed to 8 hours of estimated LED performance time, field performance ranged from 2.2 to 6.6 hours total time, which meant that the total time of biological data collection was shortened and ranged from 5.5 to 16.5 days, respectively (Table 1). Memory and power were not issues for the camera system; the 128 GB micro SD card was cleared and the battery pack was fully recharged following each deployment. Video quality was high enough to allow species-level identifications and the light from the LEDs was sufficient to light the field of view (Fig. 1F and G). The SeaBird MicroCAT-ODO also performed without any issues and had sufficient battery and memory capacity for all deployments. If not for power limitations to the LED lights, the camera system and SBE MicroCAT would have allowed for longer sampling (~1 month and potentially longer). The basic Nanolander itself can stay *in situ* for up to two years. Detailed descriptions of Nanolander performance can be found in Gallo (2018)."**

And also in the 5.0 Concluding remarks section:

**"Our deployment lengths were limited by battery capacity to power the LED lights; all other elements would have allowed for longer sampling duration. Specific ways to extend future deployment lengths are currently being explored and include: using higher efficacy LEDs, integrating additional batteries to power the LED lights into newly devised Nanolander side pods, improving circuit performance that powers the LED lights by using new camera controllers and solid-state relays, and using low-light cameras, such as the Sony 7S II, which reduce the light required to illuminate the field of view. Longer deployment lengths would be advantageous for capturing ecosystem responses to environmental variability across time-scales (hours to months)."**

Line 131: would be nice if especially details of the camera system could better show up in the improved version of Figure 1A

**We have provided the 10 page SphereCam manual as Supplement 1A.**

Line 153: please use metric units

**Changed to: "DOV BEEBE is positively buoyant in water, and is deployed with ~18 kg of sacrificial iron weights."**

Line 178: I think there is no need to use the word "high-frequency" (it's rather a matter of the perspective whether 5 min sampling rate is high-frequency or not)

**The word "high-frequency was removed". The sentence now reads:**
**"Upon recovery of the Nanolander, time-series data from the MicroCAT were analyzed to assess how environmental variability (O2, T, salinity) changes with depth."**

Line 194 please describe spiciness in a bit more detail, it's likely not common to everybody

**We have further clarified this in the text, which now reads: "Spiciness, the degree to which water is warm and salty, is a state variable that is conserved along isopycnal surfaces (Flament 2002) and can be used as a tracer for PEW (Nam et al. 2015). We calculated spiciness using the "oce" R package (Kelley and Richards 2017) and examined how oxygen concentration varies with temperature and spiciness across depths and deployments. Spiciness is used to examine**

differences in spatial variation between water masses, which otherwise may not be apparent using isopycnal surfaces because the effects of warm temperature and high salinity cancel each other out. "Spicier" water is warmer and saltier."

Line 309 deconstructed time series - please explain in more detail

**This was further clarified in the text, which now reads: "When the time series were decomposed into their additive components (i.e. daily trend, underlying trend, and random noise), time series for all depths showed a clear diurnal and semi- diurnal signal (Supplement 1E)."**

**We have also added a citation for the R package used for the decomposition in the Methods section:**
**"The oxygen time series for each deployment was also decomposed using the "stats" package (R Core Team 2019) to look at the trend, daily, and random signals that contribute to the overall data patterns."**

Figure 4: the labels for "day" and "night" are difficult to read – please enlarge

**We have enlarged the label sizes for day and night. This Figure is now Figure 6.**

Line 448: I am not sure whether the statement "At ~200 m, oxygen, temperature, and pH exhibited high variability (Fig. 2), greater at times than the variability observed at 100 m." is correct for temperature – please check.

**Thank you for pointing this out. I have taken a close look at Table 1 and inspected the overall ranges and CVs for the different deployments. For D200- DM, variability for temperature was higher than during D100-DM-Spr, based on the overall range and CV, but lower than during D100-DM-Fall. Temperature variability for D200-LJ-2 was very similar to temperature variability for D100-DM-Spr but also lower than D100-DM-Fall, based on overall range and CV (Table 1). D200-LJ-1 has lower temperature variability than both D100-DM-Spr and D100-DM-Fall.**

**I have clarified this in the results section, which reads: "While we expected that O2 variability would decrease with depth, instead we found that the greatest variability in oxygen conditions over these short time-scales was observed at ~200 m (Table 1). All three deployments from ~200 m showed broad probability density distributions of environmental conditions (Fig. 4) and large ranges in oxygen and pHest for the deployment period (Table 1). The average daily range in oxygen concentration (i.e. daily maximum-daily minimum) was highest for D200-LJ-2 (~34 µmol kg-1), followed by D200-LJ-1 (~31 µmol kg-1), followed by D200-DM (~24 µmol kg-1). The average daily oxygen range for both ~100 m deployments was lower (~20 µmol kg-1 for D100-DM-Fall and ~14 µmol kg-1 for D100-DM-Spr). The coefficient of variation (CV) for oxygen at ~200 m was twice higher than for the ~100 m deployments (Table 1). While deployments at ~300 m (D300-DM) and ~400 m (D400-DM) had much narrower probability density distributions of environmental conditions (Fig. 4), the ranges in oxygen and pHest at ~400 m were only slightly smaller than at ~300 m (Table 1). The CV for oxygen was higher at ~400 m (10.20%) compared to ~300 m (7.02%) (Table 1). The average daily range in oxygen concentration was ~11 µmol kg-1 for D300-DM and ~8 µmol kg-1 for D400-DM. Temperature did not exhibit the same pattern of variability as oxygen, with the highest variability (CV) observed during D100-DM-Fall (~100 m) (Table 1). Variability in pHest (CV) was almost twice higher at shallower depths (< 200 m), than**

**at ~300 or ~400 m (Table 1).**

Although the Figure 2 is quite attractive and informative, especially for the discussion section, when environmental variability is discussed additional Box plots might be helpful to elucidate the differences between the different deployments (i.e. depths).

**Thank you for your positive feedback on Figure 2. We have added additional violin plots for the oxygen measurements for each Nanolander deployment to a new figure (Figure 7), which additionally addresses question 4 raised by Reviewer 1. The violin plots also show the mean and +/- 1 and 2 standard deviation.**

**The Figure caption reads: "Figure 7: Comparing short-term environmental variability from *DOV BEEBE* deployments to longer-term trends using CTD casts at nearby CalCOFI station (93.3 28.0). Mean temperature (A) and oxygen (C) conditions through the water column (0-500 m) using CalCOFI CTD casts from Oct 2003-November 2019; light and dark colors indicate the variance around the mean and represent +/- 1 and 2 SD, respectively. Panels B and D show how the coefficient of variation (CV) for temperature and oxygen changes through the water column. Dotted lines in A-D indicate 100, 200, 300, and 400 m depths, and data are extracted for these depths for E-J. In E-H, violin plots show data distribution of oxygen measurements from ~16 years of CalCOFI quarterly cruises compared to ~3 week Nanolander deployments at 100 m (E), 200 m (F), 300 m (G), and 400 m (H). Violin plots show the mean +/- 1 SD (white) and +/- 2 SD (black). Panels I and J examine changes in temperature (I) and oxygen (J) conditions through time at 100, 200, 300, and 400 m. Dotted lines indicate non-significant linear relationships; solid lines indicate significant trends (p < 0.05)."**

Line 476 Turbidity can be related to local hydrodynamics caused by the energy dissipa- tion of incipient internal tides at sloping boundaries affecting the suspension, transport and deposition of food particles. If you are interested, please see e.g. Mosch et al. (2012) Factors influencing the distribution of epibenthic megafauna across the Peruvian oxygen minimum zone. Deep-Sea Research I 68 (2012) 123–135 and references therein.

**Thank you for pointing us to this interesting study. Similarly, Reviewer 1 also raised the importance of internal tides breaking on the margin as a source of environmental variability that may also explain the high turbidity conditions observed during our 300 m deployment.**

**In section 4.2 of our discussion, we have added suggested references from both reviewers. This new paragraph now reads, "The high turbidity observed at 300 m may be due to shoaling and breaking nonlinear internal waves that can form bottom nepheloid layers (McPhee-Shaw 2006, Boegman and Stastna 2019). On the Peruvian margin, energy dissipation from tidally-driven internal waves have been shown to influence the distribution of epibenthic organisms by increasing suspension, transport, and deposition of food particles (Mosch et al. 2012). High turbidity conditions have also been observed during two separate ROV dives at ~340 m off Point Loma (unpublished, NDGallo), suggesting high turbidity conditions may be the norm at these depths on the upper slope in the SCB."**

List of all relevant changes made in the manuscript:

3 new figures:

Fig. 2: A detailed schematic of the Nanolander DOV BEEBE

Fig. 3: A map showing Nanolander deployment locations with relevant bathymetry (Fig. 3)

Fig. 7: Figure comparing our short-term variability results from the Nanolander deployments with longer-term trends from CTD cast from a nearby CalCOFI station

Additional Supplementary Materials

Supplement 1A: SphereCam manual

Supplement 1C: Table of fish species observations

Supplement 1F: Additional figures added to examine relationship between tidal amplitude and oxygen variability in response to reviewer 1

Supplement 1G: Response to reviewer 1 about vertical and cross-shelf and slope structure in the water column during our deployment period.

Manuscript changes:

General edits to improve readability

Additional specificity on aspects of the nanolander that are novel to address Reviewer 2's request

Modification of Fig. 6 to increase label sizes and update of Panel C.

Addition of Section 2.3 Short-term oxygen variability in the context of longer trends (to address Reviewer 1's points about interannual variability)

Significant rewrite of Section 4.1 Comparing oxygen variability: Short-term to long-term trends to incorporate comparison to CalCOFI station data (to address Reviewer 1's points about interannual variability)

New Conclusion section (Section 5.0 Concluding remarks) added as requested by Reviewer 1, with specific additions as suggested by Reviewer 2

Additional references added as suggested by reviewers, and to further develop points the reviewers asked to be addressed

All code and data were uploaded to a Zenodo repository and the url is included in the manuscript "Code and Data availability" section. Data and code 
[revised manuscript text omitted]